RESEARCH

# Lifestyle and the presence of helminths is associated with gut microbiome composition in Cameroonians

Meagan A. Rubel[1,2,3], Arwa Abbas[4,5†], Louis J. Taylor[4†], Andrew Connell[4], Ceylan Tanes[6], Kyle Bittinger[6], Valantine N. Ndze[7,8], Julius Y. Fonsah[9], Eric Ngwang[10], André Essiane[11], Charles Fokunang[12], Alfred K. Njamnshi[13,14,15], Frederic D. Bushman[4*†] and Sarah A. Tishkoff[2,16*†]

* Correspondence: bushman@pennmedicine.upenn.edu; tishkoff@pennmedicine.upenn.edu
†Frederic D. Bushman and Sarah A. Tishkoff co-supervised this research. Arwa Abbas and Louis J. Taylor contributed equally to this work.
[4]Department of Microbiology, Perelman School of Medicine, University of Pennsylvania, Philadelphia, PA 19104, USA
[2]Department of Genetics, Perelman School of Medicine, University of Pennsylvania, Philadelphia, PA 19104, USA
Full list of author information is available at the end of the article

## Abstract

**Background:** African populations provide a unique opportunity to interrogate host-microbe co-evolution and its impact on adaptive phenotypes due to their genomic, phenotypic, and cultural diversity. We integrate gut microbiome 16S rRNA amplicon and shotgun metagenomic sequence data with quantification of pathogen burden and measures of immune parameters for 575 ethnically diverse Africans from Cameroon. Subjects followed pastoralist, agropastoralist, and hunter-gatherer lifestyles and were compared to an urban US population from Philadelphia.

**Results:** We observe significant differences in gut microbiome composition across populations that correlate with subsistence strategy and country. After these, the variable most strongly associated with gut microbiome structure in Cameroonians is the presence of gut parasites. Hunter-gatherers have high frequencies of parasites relative to agropastoralists and pastoralists. *Ascaris lumbricoides*, *Necator americanus*, *Trichuris trichiura*, and *Strongyloides stercoralis* soil-transmitted helminths ("ANTS" parasites) significantly co-occur, and increased frequency of gut parasites correlates with increased gut microbial diversity. Gut microbiome composition predicts ANTS positivity with 80% accuracy. Colonization with ANTS, in turn, is associated with elevated levels of TH1, TH2, and proinflammatory cytokines, indicating an association with multiple immune mechanisms. The unprecedented size of this dataset allowed interrogation of additional questions—for example, we find that Fulani pastoralists, who consume high levels of milk, possess an enrichment of gut bacteria that catabolize galactose, an end product of lactose metabolism, and of bacteria that metabolize lipids.

**Conclusions:** These data document associations of bacterial microbiota and eukaryotic parasites with each other and with host immune responses; each of these is further correlated with subsistence practices.

**Keywords:** Gut microbiome, Hunter-gatherers, Parasites, Helminths, Industrialization, HIV, Lactose, Diet, Metagenomic sequencing, TH2

## Background

Twenty-four percent of the world's population, predominantly in developing countries, is estimated to be infected with gastrointestinal parasites. Enteric parasites are under-studied components of the complex ecosystem of microorganisms that colonize the human gastrointestinal tract, and their effects on host physiology and the gut microbiota remain poorly understood. Industrialized countries are characterized by an overall reduction in exposure to pathogens and microbes consequent to cultural and technological societal shifts. Thus, there is a need to characterize the consequences of gastrointestinal parasite colonization and infection [1] in rural populations living traditional lifestyles to understand their effects on health [2].

The composition of the adult gut microbiome differs among ethnic groups, locations, and lifestyles [3–7]. For example, different bacterial genera dominate the fecal microbiome in people around the world living traditional rural lifestyles (*Prevotella*) compared to urban dwellers (*Bacteroides*) [4, 5, 7, 8]. Infection with gastrointestinal parasites *Entamoeba histolytica* [9], *Ascaris lumbricoides* [10], *Necator americanus* [10, 11], and *Trichuris trichiura* [10, 12, 13] also influences the structure and function of the gut microbiome. Microbiota-parasite interactions, in turn, likely influence the course of systemic infection, parasite virulence, and host immune responses [14]. Prior microbiome research has considered many gastroenteric parasites as separate exposures [9, 10, 13, 15]. However, in industrializing countries, coinfection is frequent [9], with little studied consequences. In this study, we focus on populations from Cameroon that are genetically, linguistically, phenotypically, and culturally diverse [16–20]. These populations have different types of subsistence practices but share overlapping environments and high infectious disease burdens. Here, we present a detailed analysis of sequence data from gut microbiota, quantitative measurements of multiple pathogen loads, host immune parameters, and extensive demographic data for 575 Cameroonian subjects.

## Results

In the sections below, we first present the populations studied, then quantify parasite loads. We then compare the associations of parasite burden with the gut microbiome structure, first using 16S rRNA gene tag sequencing and then using shotgun metagenomics. Lastly, we analyze a few additional features of lifestyle-microbiome interactions in the Cameroonian subjects studied.

### Populations studied

The Cameroonian populations studied were Mbororo Fulani pastoralists (hereafter referred to as "Fulani"), Baka and Bagyeli rainforest hunter-gatherers, and Bantu-speaking agropastoralists (hereafter referred to as "Bantu") (Table S1). Fulani pastoralists have subsistence practices centered largely around cattle, and the Bantu grow crops and raise livestock. The Baka and Bagyeli hunter-gatherers, who are sometimes referred to as "pygmies" for their short stature, engage in small-scale agriculture but also forage for meat and plant materials (data on diet is in Table S2- Nutritional Questionnaires, Methods). These populations were sampled over nine sites in the Northwest, South, and East regions of Cameroon (Fig. 1). As subsistence and ethnicity are strongly

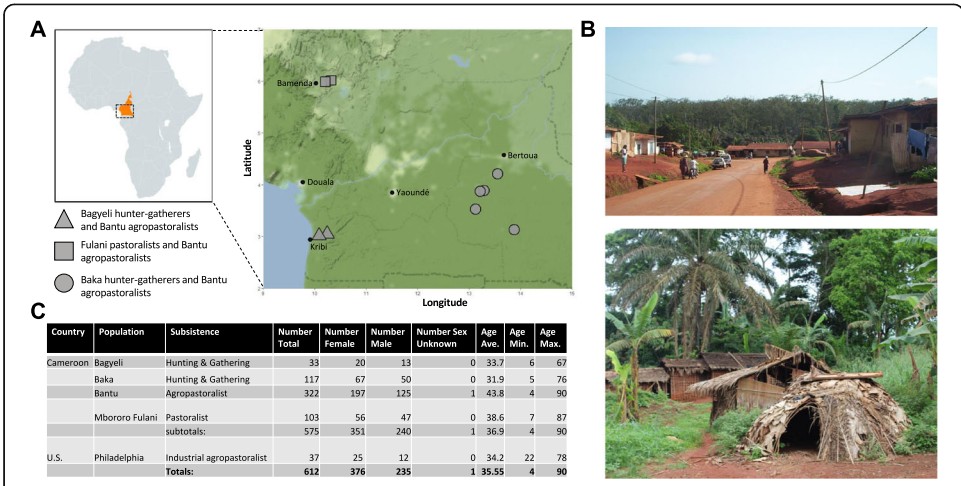

**Fig. 1** Sampling sites and demographics. **a** Cameroon sampling sites. At every sampling site, Bantu agropastoralists were sampled. One of two hunting and gathering groups (Baka and Bagyeli) or Fulani pastoralists were also sampled at these sites. **b** Top: image of a representative village with a large population of Bantu agropastoralists (creative commons license). Bottom: image of a representative village with a large population of hunter-gatherers (photograph by Meagan A. Rubel). **c** Demography table of truncated metadata for Cameroon and US (See Table S1 for full metadata)

correlated in these populations (e.g., all pastoralists are Fulani, all agropastoralists are Bantu, and all Baka and Bagyeli are hunter-gatherers), only subsistence was included as a statistical parameter in the analyses described below. We also included two cohorts of US industrial agropastoralists who have diets high in animal fats, proteins, and refined and processed foods that are the byproducts of intensive agricultural and pastoral practices, from the Human Microbiome Project [21, 22] and the COMBO study [23].

## Quantification of pathogens and their correlates with host physiology

To compare microbiome-pathogen interactions in these Cameroonian populations, we acquired data on both intestinal and blood-borne pathogens. Using thin and thick blood smears with light microscopy, a total of 198 occurrences of blood parasites were identified for *Plasmodium falciparum*, *Microfilaria loa loa*, *Mansonella perstans*, *Wuchereria bancroftii*, and *Microfilaria* spp.; *P. falciparum* accounted for the most parasites detected by microscopy ($N = 96$) (Additional file 1: Figure S1). Using a combination HIV-1/2 immunoassay from plasma, 28 HIV-positive individuals were identified.

Fecal parasites were identified by light microscopy using wet mount techniques and qPCR [24]. The concordance between microscopy and qPCR results for fecal parasite occurrence was 86% (Table S1). qPCR was more sensitive, detecting almost three times more parasite occurrences than microscopy. We tested for the following fecal parasites by qPCR: giant roundworm *Ascaris lumbricoides*, hookworms *Necator americanus* and *Ancylostoma duodenale*, whipworm *Trichuris trichiura*, roundworm *Strongyloides stercoralis*, and protists *Giardia lamblia*, *Entamoeba histolytica*, and pan-*Cryptosporidium* spp. (Additional file 1: Figure S2). None of our samples was positive for *A. duodenale*, and it was not considered in further analyses. Given the greater sensitivity, qPCR-confirmed parasite detections were used for subsequent analysis.

Counts of fecal and blood parasites indicated that Cameroonian populations had significantly different distributions of parasites (Kruskal-Wallis $\chi^2 = 244$, df = 3, $p$ value < 2.2e−16). All populations were significantly different from each other in terms of their pathogen prevalence except for the Baka and Bagyeli, who were statistically indistinguishable (FDR-corrected $p$ values < 0.05, Dunn's test of multiple comparisons). The Bagyeli hunter-gatherers had the highest individual parasite detection rate (an average of 3.91 parasites/person) compared to other Cameroonian populations (Baka = 2.83, Fulani = 0.22, Bantu = 1.13) (Fig. 2a).

Concurrent detections with > 1 tested pathogen occurred in 39% (226/575) of Cameroonians. We performed species co-occurrence analysis to identify combinations of HIV, blood, and fecal pathogens that occurred together more frequently than expected by chance. Twenty-one pathogen pairs showed significant, positive co-occurrences (FDR-corrected $p$ values < 0.05 by the hypergeometric distribution). The parasites that most frequently co-occurred were *A. lumbricoides*, *N. americanus*, *T. trichiura*, *S. stercoralis*, and *P. falciparum* ($p$ values < 0.05 by hypergeometric distribution) (Fig. 2b, Table S3). The four soil-transmitted helminth parasites are hereafter referred to as the "ANTS" group.

In addition, we looked for correlations between pathogens and a number of physiologic traits, including measures for 21 cytokines, temperature, BMI, white blood cell (WBC) count, and WBC cell type fractions (eosinophils, basophils, neutrophils, lymphocytes, monocytes; Table S1). We first determined which physiological variables were inter-correlated using correlation coefficients and tested their significance [25] (Table S4). Co-occurrence results indicated that eosinophils, white blood cell count (WBC), and temperature were significantly positively correlated with any variable that includes ANTS parasite count (e.g., ANTS binary count, ANTS count with blood parasite count) (Spearman's correlation test, corrected $p$ values < 0.01, Table S4, Additional file 1: Figure S3). Other significantly positive correlations included WBC with eosinophils and temperature (Table S4, Additional file 1: Figure S3). Eosinophilic leukocytes are normally a small fraction of WBC but increase dramatically during helminth infection [26]. No parasites were positively associated with HIV status by co-occurrence analysis.

### Association of pathogens with the gut microbiome

We amplified and sequenced the V4 region of the bacterial 16S rRNA marker gene from fecal samples obtained from 103 Fulani, 117 Baka, 33 Bagyeli, 322 Bantu, and 37 US industrial agropastoralists from the COMBO cohort [23]. A comparison of alpha diversity metrics (Faith's Phylogenetic Diversity and bacterial richness) indicated that the Bagyeli had the highest diversity and richness, and that the Fulani agropastoralists had the lowest diversity and richness, with the US being the second lowest (Additional file 1: Figure S4A-D). In all Cameroonian samples, we observed a significant positive correlation between increasing numbers of parasite detections and higher within-individual bacterial diversity (Faith's Phylogenetic Diversity) in a linear regression model (adjusted $R^2 = 0.051$, $p$ value < 0.001; Fig. 3a). When testing for correlations between bacterial diversity and ANTS count for subsistence groups, we found that the only subsistence group

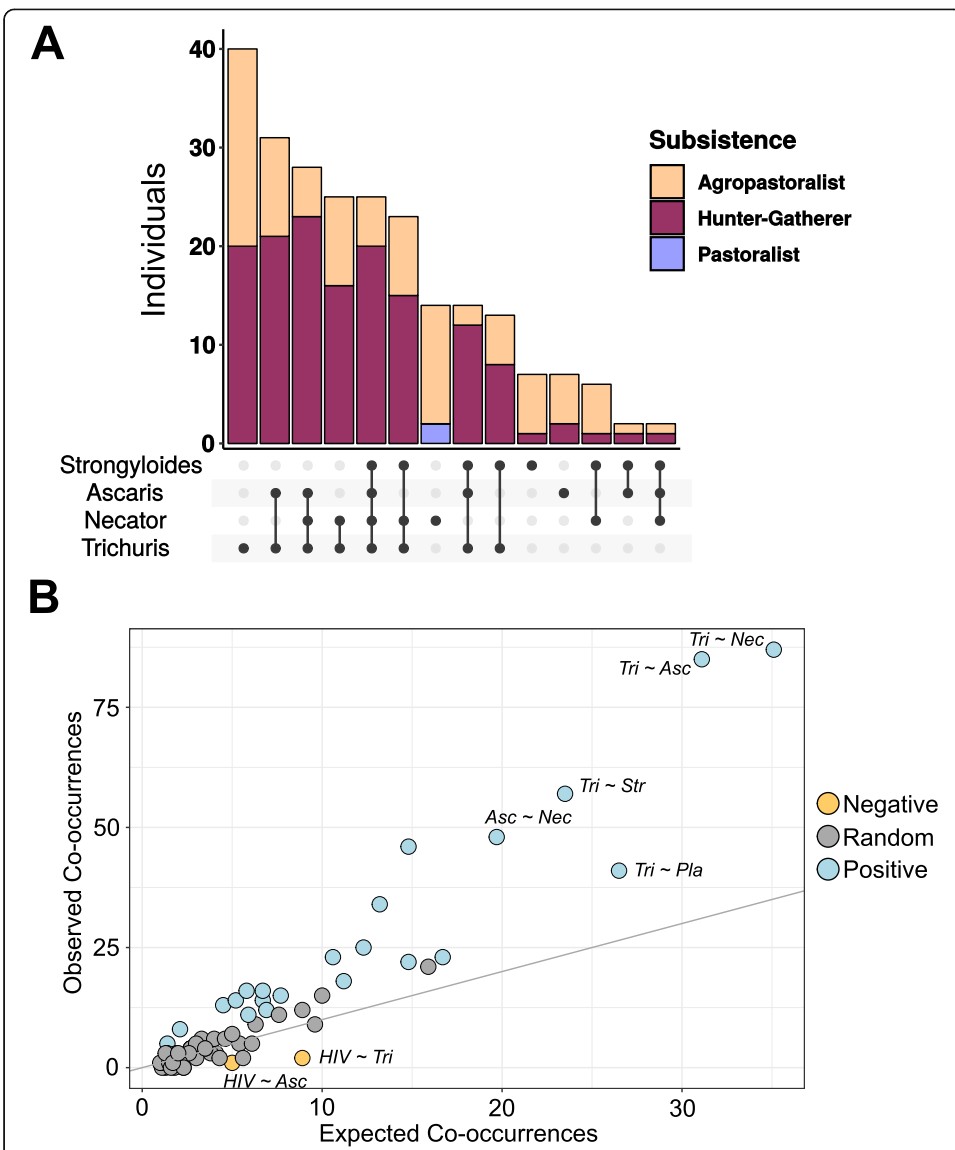

**Fig. 2** Helminth parasite characterization in the Cameroon cohort. **a** Distribution of individuals with different combinations of soil-transmitted helminths within the "ANTS" group in the Cameroon sample colored by subsistence group. **b** Positive, negative, and random association based on a probabilistic model of pathogen co-occurrence, calculated across all types of tested pathogens (blood parasites, fecal parasites, and HIV) for all Cameroonians. The expected frequency is calculated from the presence/absence data of parasites, with the assumption that the distribution of each pathogen is independent and random of other pathogens. Probabilities that are more extreme than would have been obtained by chance are shown for positive co-occurrences in blue (as one pathogen occurrence increases, the other increases), negative co-occurrences in yellow (as one pathogen occurrence increases, the other decreases), and random co-occurrences in gray (no significant association). Top hits for negative and positive co-occurring pathogens are annotated as follows: Asc, *Ascaris lumbricoides*; HIV, human immunodeficiency virus; Nec, *Necator americanus*; Pla, *Plasmodium falciparum*; Tri, *Trichuris trichiura*

that had a significant positive correlation were the Bantu agropastoralists (adjusted $R^2 = 0.018$, $p$ value $< 0.001$).

The occurrence of a single ANTS parasite was correlated with increased alpha diversity (one-way ANOVA with Tukey's post hoc testing; adjusted $p$ values $< 0.05$), paralleling the data from populations in Cameroon [9], Indonesia [10, 15], Liberia [10], Bangladesh [15],

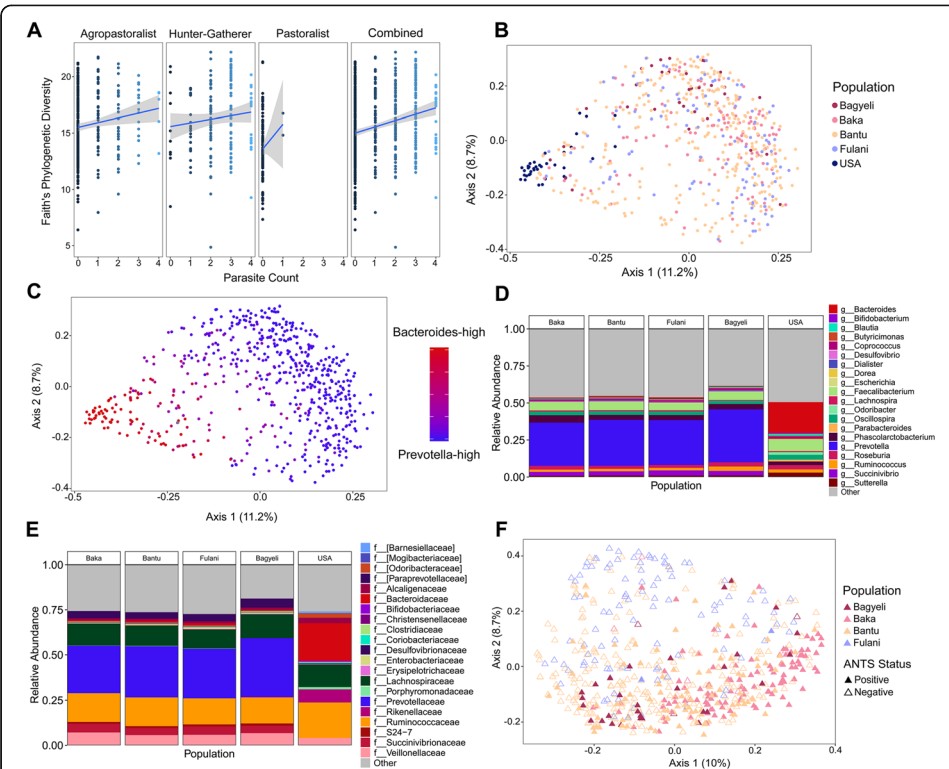

**Fig. 3** 16S rRNA diversity and composition of the gut microbiome and parasites. **a** Total bacterial diversity measured against total "ANTS" parasite count, where parasite count is the presence/absence count of unique instances of a parasite. Thus, an individual with a score of "4" has qPCR-confirmed detection for four different gastrointestinal parasites (see Table S1 for details). Bacterial diversity increases with gastrointestinal parasitemia for all Cameroonian subsistence groups. **b** Bray-Curtis distances on 16S rRNA V4 bacterial abundances show that most Cameroonians cluster separately from US individuals (PERMANOVA *p* value = 0.001). **c** 16S rRNA V4 clustering of US versus most Cameroonians along separate axes is largely reflective of the differences between two highly abundant genera, *Prevotella* and *Bacteroides*. Several pastoralists and agropastoralists overlap with the US samples. Age and sex clusters were not significant by PERMANOVA. **d** Most abundant bacterial genera per population, averaged across populations, studied in 16S analysis. **e** Most abundant bacterial families per population, averaged across populations, studied in 16S analysis. **f** Bray-Curtis on Cameroonians, showing 16S rRNA V4 bacterial abundances colored by "ANTS" parasite positivity. ANTS-positive samples are significantly different from ANTS-negative samples by PERMANOVA (*p* value = 0.001) across all Cameroonians. ANTS-positive samples remain significantly different by PERMANOVA from ANTS-negative samples when only considering Bantu agropastoralists (Additional file 1: Fig. S11B), who are the only individual population with large cohorts of both positive and negative individuals in this study (*p* value = 0.001)

and Malaysia [13]. However, this is the first time that significantly co-occurring gastro-enteric parasites have been shown to additively associate with increased bacterial diversity.

Bacterial microbiome composition in most individuals from Cameroon was significantly different from the microbiome composition of US. individuals based on weighted and unweighted UniFrac distance (Fig. 3b) (weighted UniFrac distance PERMANOVA *p*-value = 0.002, unweighted UniFrac distance PERMANOVA *p*-value = 0.003) (Fig. 3b) (Tables S5, S6). *Prevotella* was the most highly abundant bacterial genus in most Cameroonian individuals, whereas *Bacteroides* was the most abundant in the US individuals (Fig. 3c–e). We found no significant differences in weighted and unweighted UniFrac distances by sex or age using PERMANOVA. An analysis of the top ten significant metadata variables by PERMANOVA (FDR-corrected *p* values < 0.05, Tables S5, S6) revealed that parasite variables and

subsistence categories (including total ANTS and parasite counts, being positive for *Trichuris* and *Ascaris*, and subsistence) explained the most variance in gut composition (largest $R^2$ values).

ANTS-positive individuals also had significantly different bacterial composition compared to ANTS-negative individuals (Fig. 3f) (PERMANOVA $p$ values < 0.006; Tables S5, S6). Given the uneven distribution of ANTS among the different populations, we repeated the analysis considering only the Bantu agropastoralists, who had sufficient ANTS cases and controls for statistical comparison, and found a similar result (PERMANOVA $p$ values < 0.006). A possible confounding factor is that ANTS are more common in Bantu populations from the South and East than the Northwest, so we looked for differences in the gut microbiomes between Bantu individuals by region. Bacterial abundances were still significantly different based on ANTS occurrence ($p = 0.004$) in a PERMANOVA model accounting for both local geography and parasite occurrence. The majority of differences among Bantu populations were driven by taxa in the *Firmicutes* and *Bacteroidetes* phlya (Additional file 1: Figure S5A-C).

We used a supervised machine learning technique, random forest classifier (RFC), to determine which gut microbiota best predict the metadata variables identified as significantly associated with bacterial prevalence or abundance by PERMANOVA in the pooled Cameroonian dataset (Tables S5, S6). RFC analysis revealed that the country of origin could be predicted from the microbiome composition with ~ 90% accuracy (Table S7). Previous studies indicate that moderate to heavy parasite load is associated with increased morbidity [27] and could affect the gut microbiome composition. Therefore, we binned individuals based on the highest quartile of qPCR copy number for any of the four ANTS (referred to as "highly positive for ANTS"). Individuals who were highly positive for ANTS could be predicted with 83.97% accuracy. Pastoralist and hunter-gatherer subsistence, as well as individual presence of *Microfilaria* spp., *A. lumbricoides*, or *T. trichuris*, could be predicted with 81–82% accuracy. Finally, positivity for any ANTS parasite could be predicted with ~81% accuracy (Table S7).

To determine the importance of a given taxon predicted by RFC for a particular variable, we plotted the proportional abundance of the top ten taxa from the RFC analysis. This revealed that hunter-gatherers have statistically significantly higher abundances of *Bacteroidales*, *Prevotella stercorea*, *Succinivibrio*, *Phascolarctobacterium*, and *Treponema*; pastoralists have higher abundances of *Odoribacter*, *Rikenellaceae*, *Bacteroides caccae*, and *Bacteroides ovatus*; and agropastoralists have comparatively lower abundances for all of these taxa (Fig. 4a; Table S7). *Bacteroidales*, *CF231*, *Treponema*, *Prevotella stercorea*, *Anaerovibrio*, and *Succinivibrio* were statistically significantly increased with ANTS positivity (Wilcoxon rank-sum test with continuity correction $p$ values < 0.05; Additional file 1: Figure S6; Table S7). In addition, we analyzed the Bantu agropastoralists separately, to avoid confounding with subsistence. In this subset of individuals, we found that the microbial taxa most associated with ANTS positivity were *CF231*, *Bacteroidales*, *Succinivibrio*, *Treponema*, and *Clostridiaceae* (Fig. 4b). From these analyses, we find that the only bacteria predictive solely of ANTS occurrence and not also predictive of subsistence were *Ruminococcus bromii* (increased abundance in ANTS-negative individuals) and *CF231* (increased abundance in ANTS-positive individuals).

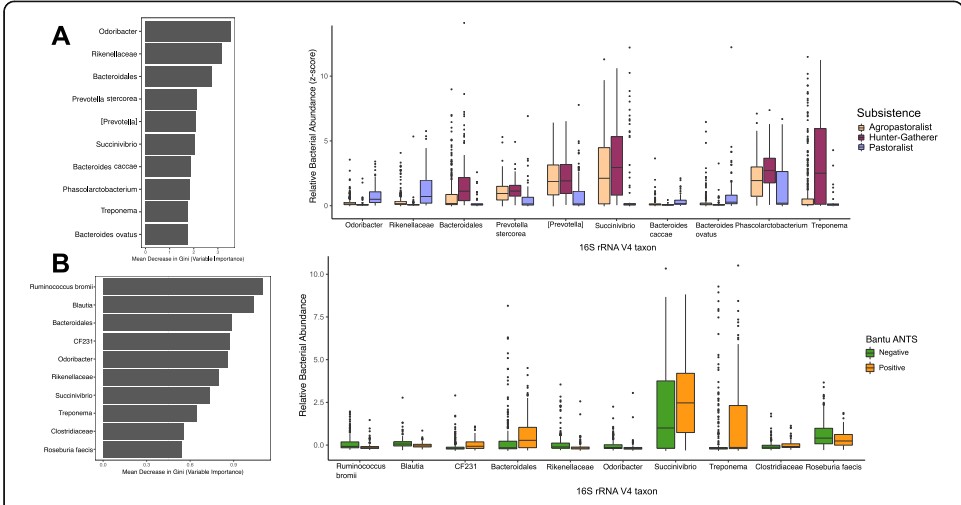

**Fig. 4** RFC on subsistence (all Cameroonians) and ANTS occurrence in Bantu. **a** Top ten most important taxa in predicting Cameroonian subsistence in 16S rRNA data, shown by model importance on the left and relative bacterial abundance (*z*-scores) on the right. **b** Top ten most important taxa in predicting ANTS occurrence in Bantu agropastoralists only in 16S rRNA data, shown by model importance on the left and relative bacterial abundance (*z*-scores) on the right

### Association of microbiome structure and blood-borne pathogens

*P. falciparum* infection also correlated with gut microbiome composition and could be predicted from the gut microbiome composition with ~ 73% accuracy. The top five taxa that best predicted *P. falciparum* infection status were *Bacteroidales*, *Roseburia faecis*, *Lachnospiraceae*, *Coprococcus*, and *Desulfovibrio* (Table S7). However, *P. falciparum* detection explained less variance in the gut composition in PERMANOVA analysis than did ANTS (Tables S5, S6), so we focused on the ANTS in most of the analysis here.

Twenty-eight of the subjects were found to be HIV-positive. Comparison of the members of this group to HIV-negative individuals showed no distinction based on weighted and unweighted UniFrac distances in PERMANOVA analysis (Tables S5, S6).

### Relationship between ANTS infection and the microbiome using shotgun metagenomic sequencing

We performed shotgun metagenomic sequencing on a subset of 175 Cameroonian fecal samples (94 Bantu, 37 Baka, 22 Bagyeli, 22 Fulani) to investigate the association of both bacterial and non-bacterial members of the microbiome and ANTS detection and to assess the associations between ANTS detection and gene function. These samples were selected to include ANTS-positive and ANTS-negative individuals and diverse subsistence groups. Where possible, ANTS-positive subjects and controls were matched by sex, age, and sampling site (Table S1). The Cameroonian cohort was compared to 27 healthy US human gut microbiome samples from the HMP cohort [21, 22].

Alpha diversity (Shannon Index) and evenness (Simpson's Index) results derived from shotgun metagenomic sequencing data paralleled the 16S rRNA marker gene sequencing (Additional file 1: Figure S7). The Fulani pastoralists were the only population that had significantly different microbial diversity from all other Cameroonian populations

(1-way ANOVA with Tukey's honest significant difference test, adjusted *p* values < 0.05) and had comparatively reduced alpha diversity by both metrics. Samples from healthy US urban individuals from the HMP cohort had significantly lower alpha diversity than the Fulani population (*p* = 0.035 1-way ANOVA with Tukey's honest significant difference test on Simpson Index, *p* = 0.035).

Metagenomic reads were pre-processed using the Sunbeam pipeline [28] and assigned to microbial taxa using several methods, including MetaPhlAn2 [29], KrakenUniq [30], and alignment to the Greengenes reference database of 16S gene sequences [31, 32]. The top three bacterial genera within the Cameroonian cohort using the metagenomics databases were *Prevotella*, *Bacteroides*, and *Faecalibacterium* (Fig. 5a). Shotgun metagenomic sequencing also showed that the Fulani pastoralists had higher levels of *Bacteroides* (median proportional abundance 29%) and lower levels of *Prevotella* (median proportional abundance 1%) than other Cameroonian populations, as was observed by 16S gene sequencing (Additional file 1: Figure S8). Fulani and US samples shared high relative bacterial abundances of *Bacteroides* (median 59% US, median 29% Fulani) and *Alistipes* (median 4% US, median 3% Fulani) compared to the other Cameroon populations (Fig. 5b). As an additional point of comparison, we queried our reads against a representative set of ~ 5,000 metagenome-assembled genomes (MAGs) from Pasolli et al. [33] (Additional file 1: Figure S9) and found that the results of this analysis were concordant with those above.

When identifying sequences using the KrakenUniq database, we found that a larger fraction of metagenomic sequences remained unclassified in Cameroonian fecal samples compared to those from US urban dwellers in the HMP dataset. Among the Cameroonian groups, the Fulani had the lowest proportion of unclassified sequences, which

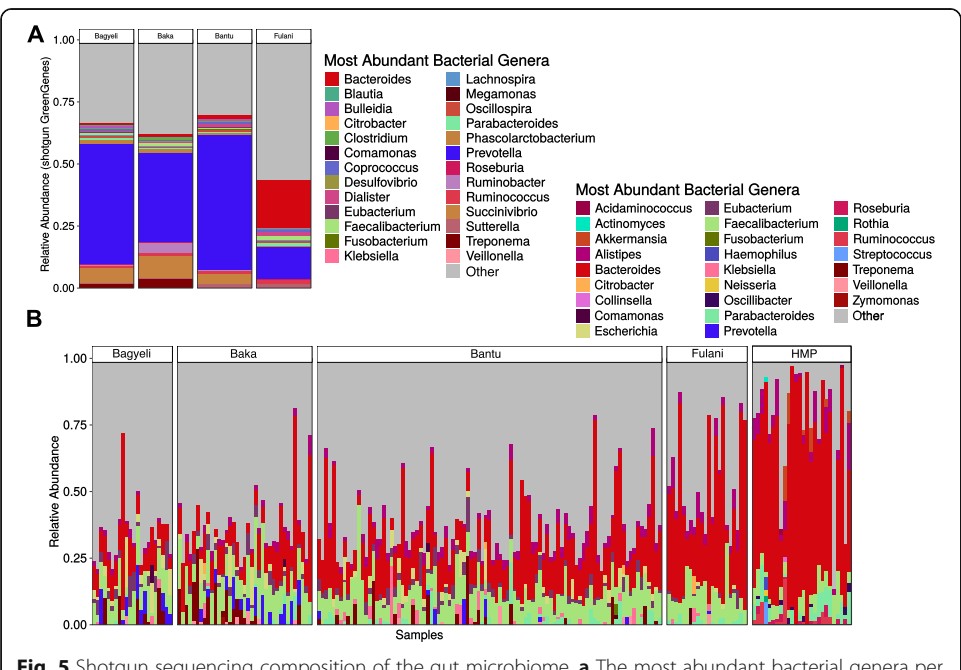

**Fig. 5** Shotgun sequencing composition of the gut microbiome. **a** The most abundant bacterial genera per population, averaged across populations, studied in shotgun metagenomics analysis. **b** The most abundant bacterial genera per population studied in shotgun metagenomics analysis. Individual vertical bars represent different samples. Human Microbiome Project (HMP) samples are included for comparison

is likely attributable in part to their high *Bacteroides* abundance, members of which are well-studied and well-represented in genomic sequence databases [33]. The Baka had the greatest number of reads unclassifiable at any level.

Reads classified as viral, fungal, and parasitic-eukaryotic represented a small fraction of the average total reads across samples (average 0.05%, max 1.5%). Although samples were not purified for virus-like particles, we occasionally detected eukaryotic viruses (Additional file 1: Figure S10), notably human mastadenovirus D. Human mastadeno-virus D species have associations with gastrointestinal, respiratory, and eye infections and have been found in river and drinking water in South Africa [34] and could be in Cameroonian water sources. However, we did not find significant differences between subsistence groups, sampling sites, and regions in frequency of human mastadenovirus D (Fisher's exact test with Bonferroni test correction, $p$ values > 0.05). The detection of other eukaryotic viruses was too infrequent to test for statistically significant differences between populations.

Metagenomic samples from four Cameroonian populations significantly clustered by Bray-Curtis dissimilarities based on ANTS positivity (PERMANOVA $p$ values < 0.05) (Additional file 1: Figure S11A). ANTS-positive Bantu agropastoralists were signifi-cantly different from ANTS-negative Bantu agropastoralists (first principal component tested, Wilcoxon rank-sum test with continuity correction $p$ value = 0.008) (Add-itional file 1: Figure S11B); as mentioned above, the Bantu are the only group with ad-equate cases and healthy controls for within-group ANTS statistical tests.

As a check on data quality, we compared ANTS detection in the metagenomic data versus qPCR. There was a significant positive correlation between molecular (qPCR cycle of threshold) and metagenomic detection (total $k$-mers) of *A. lumbricoides* ($p$ value < 0.001, Spearman's $\rho = -0.74$), *N. americanus* ($p$ value < 0.001, Spearman's $\rho = -0.63$ and *T. trichiura* ($p$ value = 0.0008, Spearman's $\rho = -0.36$) (Fig. 6a). We used unique $k$-mers (called by KrakenUniq) as a measure of genome coverage for this ana-lysis rather than reads, as unique $k$-mer counts are robust to potentially spurious read pileups over short genomic regions. Genome size was also a factor—we detected para-sites with large genomes (*N. americanus*, *A. lumbricoides*) more efficiently than para-sites with smaller genomes (*T. trichiura*, *Cryptosporidium* spp.). For *Cryptosporidium* spp. and *S. stercoralis*, a positive trend was observed between molecular and metage-nomics detection.

### Analysis of additional eukaryotic gut organisms

By analyzing the shotgun metagenomic data, we were able to identify potential presence of parasite species other than those identified by qPCR. We identified different species of *Entamoeba* in the shotgun analysis which were not detected by species-specific qPCR for *Entamoeba histolytica*. Of those known to infect humans [35], *Entamoeba dispar*, *Ent-amoeba coli*, and *Entamoeba hartmanni* were co-detected with *E. histolytica* in 33 sam-ples (Fig. 6b). With the exception of one Fulani individual who was positive for both *E. hartmanii* and *E. histolytica*, all other *Entamoeba* detections occurred in agropastoralists and hunter-gatherers. Given the different species of *Entamoeba* detected in shotgun se-quencing, three additional classification RFCs were run to test whether the gut microbiota composition could predict positivity for commensal *Entamoeba* (*E. coli*, *E. dispar*, *E.*

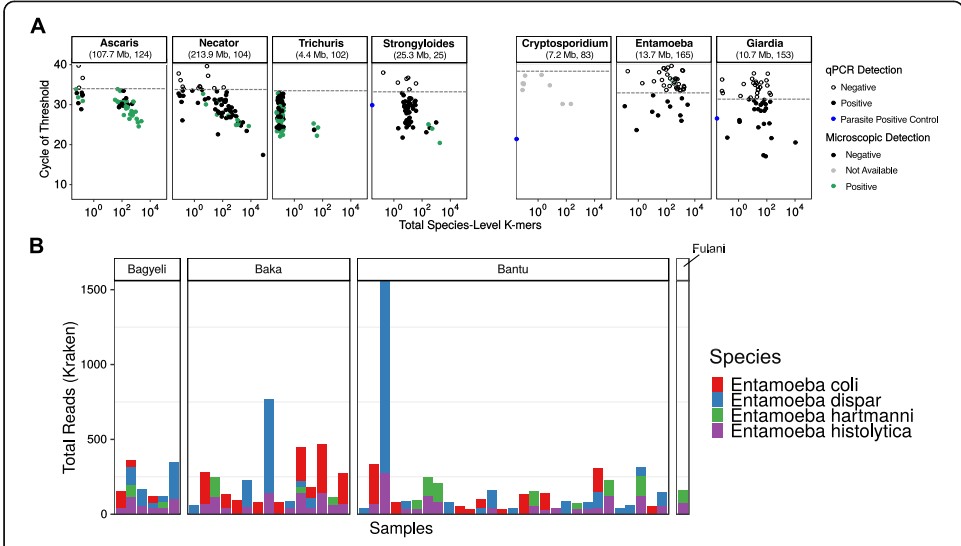

**Fig. 6** Comparisons of parasite detections using qPCR and shotgun sequencing. **a** Comparison of average Ct (cycle threshold) values are shown along the *y*-axis for ANTS (to the left of **a**), *Cryptosporidium* spp., *Entamoeba histolytica*, and *Giardia lamblia*. The *x*-axis corresponds to log10-transformed total *k*-mers from shotgun metagenomic sequencing. A lower average Ct translates to an earlier cycle fluorescence threshold and higher initial parasite genomic copy number. Lower Ct values have higher total *k*-mer counts and correspond to a higher copy of parasite genomes. The dashed line represents the Ct value of the highest standard. Blue dots along the *y*-axis represent positive controls, which were not shotgun sequenced. Filled dots are samples that were positive in qPCR detection, and green filled dots were simultaneously positive in fecal microscopy. **b** Metagenomic read counts in samples that were qPCR tested for *Entamoeba histolytica* only; however, given the amount of hits to qPCR negative targets, we only looked at reads assigned to species of *Entamoeba* known to infect humans

*hartmanni*), pathogenic *Entamoeba* (*E. histolytica*), or any *Entamoeba*, which had 69%, 54%, and 72% model accuracy, respectively. Of these, eight taxa had significantly different abundances between *Entamoeba*-negative and *Entamoeba*-positive categories (FDR-corrected Wilcoxon rank-sum test *p* values < 0.05). We note that assignment of multiple species is complex and could be confounded by the presence of new species with genome sequence intermediate between known species.

Bacterial taxa that had significantly higher abundances in *Entamoeba*-negative (Ent-) individuals were *Flavobacterium magnum*, *Shigella dysenteriae* (*S. dysenteriae*), and *Anoxybacillus kamchatkensis*. The taxa that had significantly higher abundances in *Entamoeba* positive (Ent+) individuals were *Erysipelotrichaceae*, *Trueperella pyogenes*, *Staphylococcus aureus* (*S. aureus*), and *Blastocystis hominis* (*B. hominis*). Members of the *Erysipelotrichaceae* family have been associated specifically with *Entamoeba* infection in western lowland gorillas [36] and in humans [10]. Both *S. aureus* (associated with Ent+) and *S. dysenteriae* (associated with Ent-) can induce changes in *E. histolytica* virulence and host response through the modification of *E. histolytica* surface lectin expression, adhesion, cytotoxicity, and proteolysis [37]. *Trueperella pyogenes* (Ent+), *Flavobacterium magnum* (Ent-), and *Anoxybacillus kamchatkensis* (Ent-) have not been associated before, to our knowledge, with *Entamoeba* positivity.

Finally, *B. hominis* is a unicellular protozoan found in human large intestines and stool at rates higher than any other parasite in non-industrialized countries [38, 39]. Although *B. hominis* is usually considered a non-pathogenic commensal, and we had not

detected it with microscopy, *B. hominis* has been noted to associate with increased diversity of human gut bacteria [40].

## Features of subsistence groups in metagenomic data

Random forest classifiers using shotgun data were less accurate than 16S RFC models at the prediction for subsistence group (64% accuracy vs. 72%), but still detected three genus-level and species-level taxa that matched those identified in the 16S marker gene analysis: *Bacteroidales bacterium* CF, *Phascolarctobacterium succinatutens*, *Treponema succinifaciens*, and *Bacteroides caccae* (Additional file 1: Figure S12A). ANTS positivity was again a strong predictor of microbiome composition (77% accuracy) among the tested metadata variables in shotgun RFC classification (Additional file 1: Figure S12B) and had higher accuracy than the RFC for Bantu individuals who were ANTS-positive (RFC 59% accuracy) (Additional file 1: Figure S12C). Two taxa that predicted ANTS positivity independent of subsistence were *Peptoclostridium acidaminophilum* and *Candidatus Azobacteroides pseudotrichonymphae*. Both taxa had significantly higher abundances in ANTS-positive individuals in the shotgun cohort and within Bantu-only (two-tailed Wilcoxon rank-sum test, $p$ values < 0.05).

## Association of ANTS parasite detection and bacterial gene content

To begin to assess functional interactions between ANTS and bacteria, shotgun reads were annotated using the Kyoto Encyclopedia of Genes and Genomes (KEGG) [41, 42] using DIAMOND [43]. Pathways affected were identified using linear discriminant analysis effect size (LefSe) [44]. Several KEGG classes were significantly differentially enriched among specific subsistence groups (Fig. 7a; Table S8). In agropastoralists, we found an enrichment of bacterial gene pathways involved in streptomycin biosynthesis, acarbose and validamycin biosynthesis, beta-lactam resistance, and cationic antimicrobial peptide (CAMP) resistance. These gene pathways are all involved in the production of antibiotics or antibiotic resistance. Hunter-gatherers had an enrichment of genetic information processing (GIP) pathways (e.g., aminoacyl tRNA biosynthesis, RNA polymerase) and microbially mediated disease pathways in addition to enrichment in methane, purine, and pyrimidine metabolism. The Fulani had enrichment for pathways involved in galactose, starch and sucrose, glycan, and lipid metabolism (glycosphingolipid, glycerolipid, glycerophospholipid, and sphingolipid metabolism and glycosphingolipid biosynthesis; Fig. 7a).

We again considered Bantu ANTS-positive and ANTS-negative individuals in a separate analysis, but found no significantly different pathways after multiple test correction. Across the entire Cameroon cohort, ANTS-negative individuals tended to have pathway differentiation that closely followed the results for subsistence, which is likely a reflection of the comparatively larger populations of ANTS-positive individuals in hunter-gatherer and agropastoralist groups versus pastoralists (Fig. 7b). However, ANTS-positive individuals had enrichment for genes that play a role in bacterial purine and pyrimidine metabolism as well as nutrient signaling pathways implicated in aging (also known as the "longevity regulating pathways"). Auxotrophic parasites that are deficient in purines and pyrimidines must salvage these nucleotides from extraneous sources to synthesize DNA for their survival

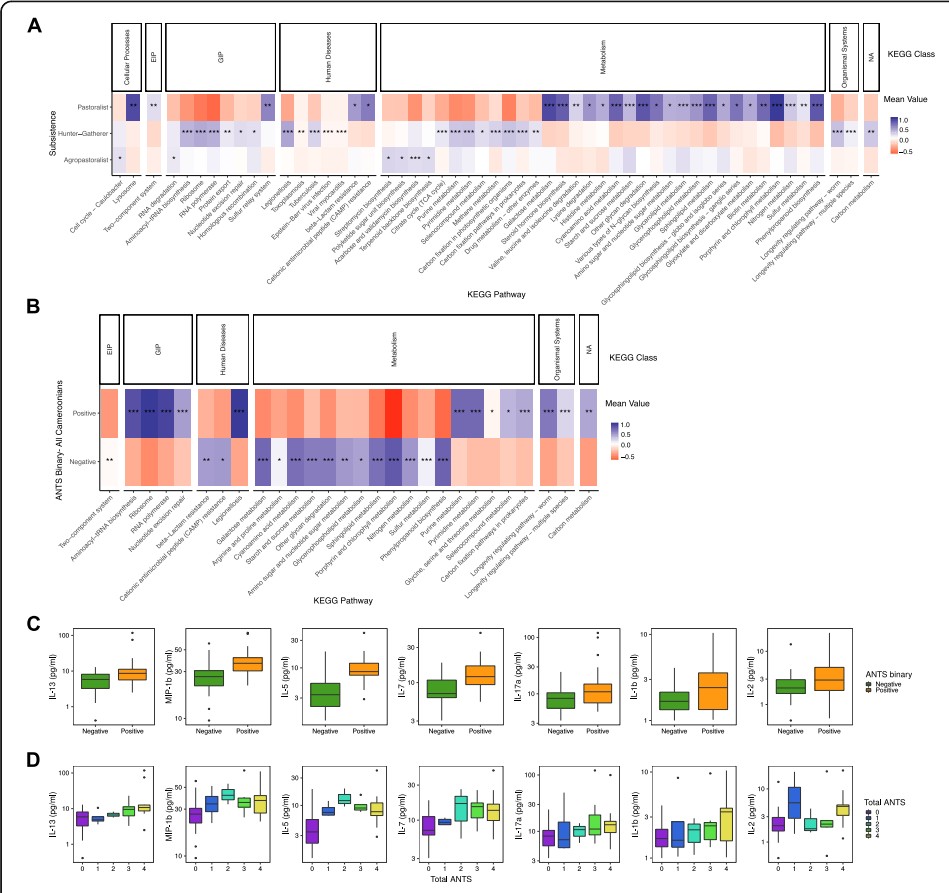

**Fig. 7** Functional analysis of gene content by pathway and association of ANTS with cytokine levels. **a** Functional analysis of gene content by pathway for subsistence using KEGG. Each row represents a subsistence group, and each column represents a gene pathway. Columns are grouped according to the KEGG class. The intensity in each cell represents the mean value of relative enrichment, with dark purple representing high pathway enrichment and dark red representing low pathway enrichment. Cells marked with asterisks denote significance, with *$p = 0.05$, **$p = 0.001$, and ***$p < 0.001$. EIP, environmental information processing; GIP, genetic information processing. **b** Functional analysis of gene content by pathway for Cameroonians positive or negative for ANTS using KEGG. Each row represents a subsistence group, and each column represents a gene pathway. Columns are grouped according to the KEGG class. The intensity in each cell represents the mean value of relative enrichment, with dark purple representing high pathway enrichment and dark red representing low pathway enrichment. Cells marked with asterisks denote significance, with *$p = 0.05$, **$p = 0.001$, and ***$p < 0.001$. EIP, environmental information processing; GIP, genetic information processing. **c** Boxplots showing the levels of all cytokines determined as positively correlated with ANTS parasites (Tables S10, S11), binned by positive/negative status. **d** Boxplots showing the same cytokines indicated as significantly correlated from **c** but compared across counts of unique ANTS (e.g., a dot in the "4" boxplot is an individual who was positive for all four ANTS). Individuals are counted only once

and proliferation [45]. Purine-salvaging parasites that were found in the Cameroonians included the protozoans *Giardia* spp., *Plasmodia* spp., *Entamoeba* spp., and *Cryptosporidium* spp., as well as the nematode *S. stercoralis* [46]. We assessed bacteria contributing to the enrichment of KEGG pathways in ANTS-positive individuals and identified a 100% overlap between these bacteria and the top ten taxa that were most predictive of ANTS positivity in RFC analyses using shotgun sequencing data (Table S9). This result serves as a cross validation that these taxa and their functional pathways are significantly different in ANTS-positive samples.

### Comparison of microbial data to human immune responses reported by cytokine production

The concentrations of 19 cytokines across 72 Cameroonian plasma samples were measured and visualized in separate correlation plots by ANTS detection ("ANTS binary"), ANTS total count (where count is 0–4), HIV (presence/absence), and *P. falciparum* (presence/absence). ANTS detection was significantly positively associated with Th1-associated cytokines IL-7 and IL-2, Th2 cytokines IL-5 and IL-13, Th17 cytokine IL-17a, and proinflammatory cytokines MIP1b/CCL3 and IL-1b (Sup. Fig. 12A; Fig. 7c) (Spearman's correlation coefficient, *p* values < 0.01, Table S10). The count of ANTS was positively associated with the same cytokines, except IL-2 (Additional file 1: Figure S13B; Fig. 7d) (Spearman's correlation coefficient, *p* values < 0.01, Table S11). Levels of cytokines IL-13, IL-17a, and IL-1b increased with ANTS count, whereas cytokine values peaked at an ANTS count of two for MIP-1b, IL-5, and IL-7. HIV and *P. falciparum* had positive, significant associations with proinflammatory cytokine TNFα (Spearman's correlation coefficient, *p* values < 0.01, Additional file 1: Figure S13C, Table S12) and IL-10 (Spearman's correlation coefficient, *p* values < 0.01, Additional file 1: Figure S13D, Table S13), respectively.

Regression-based RFC was performed on all cytokines in conjunction with 16S gene and shotgun metagenomic data to assess whether the microbiome composition could be used to predict cytokine values (Table S6, Additional file 1: Figure S14). Regression-based RFCs indicated that microbiome data could most accurately predict IL-5 levels among all tested cytokines (16S rRNA sequencing data, 31% accuracy; shotgun sequencing data, 75% accuracy). IL-5 is essential in the development and recruitment of eosinophils to sites of infection and stimulates the production of anti-microbial peptides and mucus in the intestinal epithelium during helminthiasis [47, 48]. The higher accuracy of shotgun sequencing data in predicting IL-5 levels as compared to 16S gene sequencing data is likely the result of a larger percentage of ANTS-positive individuals in the shotgun sequencing dataset, as IL-5 is positively associated with helminthiasis (Fig. 7c, d; Sup. Figure 12A, B). Of note, the unassigned bacteria within the *Bacteroidales* class were most predictive of IL-5 levels based on shotgun sequence data and were the second most predictive class of bacteria using the 16S sequencing data (Sup Table 6). In both datasets, *Bacteroidales* abundance had a significant, positive relationship with IL-5 levels (Spearman's test for correlation in 16S *p* value < 0.001; ANOVA in shotgun *p* value < 0.001). *Bacteroidales* was also the most explanatory taxon in highly positive ANTS and ANTS detection ("ANTS binary") using shotgun and 16S sequence datasets. *Bacteroidales bacterium* CF had significantly higher abundance in ANTS-positive versus ANTS-negative individuals (Wilcoxon rank-sum test with continuity correction, *p* value < 0.001) using both shotgun and 16S sequencing data (*p* value < 0.001).

### Further analysis of gut microbiota, subsistence strategy, and diet: gut microbiota are not associated with the lactase persistence phenotype in Cameroonians

The ability to digest milk as an adult is suggested to be an adaptive trait that can confer nutritional benefits and can provide a critical source of water in arid regions [49]. Prior gut microbiome studies have contrasted industrialized populations with hunter-gatherer or agropastoralist populations, and only one has included African pastoralists

known to consume high volumes of dairy [50]. Here, we compared functional measures of the ability to breakdown milk sugars with microbiome data for Cameroonians, which includes a pastoralist population.

The ability to break down lactose milk sugar in the small intestine past weaning and into adulthood is known as the lactase persistence (LP) phenotype [51]. We predicted that hosts who are lactase non-persistent (LNP) would not be capable of metabolizing lactose in the small intestine, and instead, the lactose would be catabolized by bacteria containing the LAC operon in the distal colon. The LAC operon, which produces beta-galactosidase, cleaves the milk sugar lactose into galactose and glucose, and gut bacteria with this operon may play a role in the digestion of dairy products (Additional file 1: Figure S15).

We administered a lactose-tolerance test (LTT) to 154 Cameroonians to test for the association of the gut microbiome with the LP/LNP phenotype. Of the Cameroonians, 102 were LNP, 21 were LP, and 31 were lactose intermediate-persistent (LIP). The latter two groups were both considered "positive" in LP binary analysis, where samples were considered as either lactase-persistent (LP = 52) or non-persistent (LNP = 102) (see the "Methods" section, Table S1). Bacterial microbiome composition (16S) across the entire Cameroonian cohort did not differ by lactase persistence phenotype in weighted or unweighted PERMANOVA (Tables S4, S5). In addition, microbiome composition was not a strong predictor of lactase persistence phenotype in either 16S (65.56% accuracy) or shotgun data (48.84% accuracy).

However, we found an enrichment of genes in the galactose and lipid metabolism pathways for Fulani pastoralists compared to other populations using shotgun sequencing data (Fig. 7a). The most highly abundant bacteria contributing to genes in these enriched pathways were *Bacteroides vulgatus* for galactose metabolism, and *Prevotella enoeca*, *Bacteroides fragilis*, and *Bacteroides vulgatus* for lipid metabolism.

## Discussion

Here, we present a large-scale study of the microbiome of 575 Cameroonians, interrogated using both shotgun and shotgun metagenomic and 16S rRNA amplicon sequencing, and compare the results to data on pathogen load, anthropometrics, immune parameters, and lactose tolerance. Several previous studies have provided data on the gut microbiota of rural populations with traditional subsistence strategies [3, 4, 8, 33, 52–57], and two compared data on parasite infection in Africans [9, 10]. Our data provide detailed insights thanks to a greater power from the larger sample set, allowing, for example, the finding that presence of multiple parasites showed relatively stronger effects on the gut microbiota. We found that the occurrence of *Plasmodium* had a detectable effect on the gut microbiome, as in previous studies [58, 59], but the presence of ANTS had a much larger effect, and so, ANTS are the focus here. Below, we discuss first the features of the gut microbiota newly seen here, then the interaction with ANTS; lastly, we discuss a few additional findings on subsistence strategy and microbiome structure.

We show that Cameroonians have higher amounts of *Prevotella* relative to *Bacteroides*-enriched US samples. *Bacteroides* relative abundance is higher in Fulani than in other Cameroonian populations. Our ability to identify microbes from remote, rural settings is limited by the availability of appropriate reference microbial genomes. As a

result, we have a larger fraction of "unknown" reads that could not be classified at any taxonomic level in Cameroonian samples as compared to US samples. Our sample set also allows us to ask how the different baseline microbiotas in rural populations are associated with colonization with multiple parasites.

We show that increasing parasite count correlated with higher alpha bacterial diversity. We speculate that this higher diversity associated with helminth positivity could result from direct or indirect communication with other microbes and/or with immunomodulatory mechanisms (i.e., cytokines). Increased gut bacterial diversity has been both positively and negatively associated with intestinal helminth and protist infections [14].

Two studies, Gomez et al. [4] and Morton et al. [9], have previously characterized the bacterial gut microbiomes of some Central African rainforest hunter-gatherer populations and neighboring Bantu agropastoralists. Gomez et al. [4] hypothesized that enrichment of *Firmicutes* in the Bantu as compared to the Baka may be associated with an increased ability to catabolize dietary energy. In contrast to this, we did not find an enrichment of *Firmicutes* in the Bantu as compared to the Baka. From 16S rRNA sequencing, we found that the median proportional abundance for *Firmicutes* to *Bacteroidetes* was approximately 1:1 for both the Baka and the Bantu. The difference in the *Firmicutes* to *Bacteroidetes* ratios between our study and Gomez et al. [4] may be due in part to several factors, including distance from the sampling location in Gomez et al. [4] to our nearest sampling location (> 350 km apart) diet, bias in assessment of microbial abundances introduced by different primers and amplicons between studies, and the comparatively larger number of individuals sampled in our study (117 Baka and 322 Bantu) compared to Gomez et al. [4] (28 Baka and 29 Bantu).

Morton et al. [9] characterized the bacterial microbiome of rural populations in Southwest Cameroon that were highly parasitized by protists and helminths through 16S rRNA gene sequencing. Morton and colleagues found that *E. histolytica* colonization was predictive of microbiome diversity and composition (79% accuracy) [9]. Here, we found that *A. lumbricoides*, *N. americanus*, *T. trichiura*, and *S. stercoralis* (ANTS) occurrence is highly predictive of microbiome composition, particularly when more than one species is present at the same time (81% accuracy). Microbiome composition could be used to predict ANTS parasite co-occurrence with greater accuracy than all other tested variables save country and subsistence, indicating the importance of considering gut parasite co-occurrence when studying microbiome composition in populations known to have helminthiasis.

Shotgun sequencing suggested that several Cameroonians harbored multiple types of commensal and pathogenic *Entamoeba*. Co-occurrence of commensal and pathogenic *Entamoeba* species has been previously documented in human and non-human primate hosts in east Cameroon [60]. The degree to which human commensal and pathogenic enteric *Entamoeba* interact, enhance, or inhibit one another remains an open area of investigation.

Multiple RFC models testing different categories of ANTS parasite detection indicated that *Bacteroidales*, a known occupant of intestinal mucosal surfaces with mucin degrading species, was an important predictive taxon. *Bacteroidales* was consistently found at elevated abundances in ANTS-positive individuals. Previously, *Bacteroidales* has been found in lower abundances in the guts of humans infected with *Entamoeba*

*histolytica* [9]. In murine models, infection with helminths led to the reduction of *Bacteroidales* and the concurrent expansion of *Clostridiales* communities [61], which was hypothesized to stimulate an anti-inflammatory response (increased IL-5 and IL-13) in the host. *Bacteroidales* have been shown in multiple studies to modulate intestinal and immune functions in the host [62–64]; here, *Bacteroidales* may be found in higher abundances due to the unexplored direct interactions with ANTS, through indirect means via immunomodulatory signaling as a result of ANTS positivity, or due to an increase in mucin production consequent to nematode-induced morbidity [65]. Our observations are consistent with similar *Bacteroidales* expansions detected in the gut microbiomes of rural Malaysians infected with *T. trichiura* [13]; however, we cannot rule out that *Bacteroidales* abundances found here may be confounded with subsistence practice.

RFC results suggested that four taxa, *Ruminococcus bromii*, *CF231*, *Peptoclostridium acidaminophilum*, and *Candidatus Azobacteroides pseudotrichonymphae*, were associated with ANTS detection after correction for subsistence strategy. *CF231* is a common occupant of ruminant guts [66] which has occasionally been found in humans [67] but has not, to our knowledge, been associated with human fecal parasites. *Ruminococcus bromii* can degrade foods high in resistant starch and has been associated with fishing subsistence in coastal Cameroonians [9]. In our study, we found that individuals with this bacterium were less likely to be ANTS-positive even when controlling for subsistence. Diets rich in resistant starch have been linked to reductions in gastrointestinal inflammation [68]; thus, it is possible that *R. bromii* could have a protective effect against helminthic disease, or at least in alleviating inflammation associated with helminthic disease. *Peptoclostridium acidaminophilum*, previously known as *Eubacterium acidaminophilum* [69], is a versatile, amino acid-degrading anaerobe that has not been associated, to our knowledge, with helminthiasis in prior research. However, Hadza hunter-gatherers were previously described to have an enrichment of KEGG orthologous genes involved in amino acid metabolism, and greater functional potential to metabolize proline, serine, glycine, and threonine [56]. *Candidatus Azobacteroides pseudotrichonymphae* is a termite endosymbiont [70]. These bacteria, together with the taxa in the carbohydrate-metabolizing *Ruminococcaceae* family and non-pathogenic species in the genus *Treponema* (all of which are present in our data), are considered common occupants of termite guts [71]. *Treponema* is also a common constituent of healthy non-human primate guts [72, 73]. Termites are a substantial component of hunter-gatherer and agropastoralist diets in many parts of Cameroon [74, 75]. Termite consumption could be more common in ANTS-positive individuals, which could be affected by bioavailability of termites, subject to climate and location (i.e., more tropical locations, which are correlated with a higher infectious disease burden of helminths). Whether the presence of these taxa has any effect on susceptibility or response to ANTS parasites remains an open question.

We found no detectable microbial community alterations in HIV+ compared to HIV– subjects (Sup. Figures. 15 and 16), paralleling some, but not all, studies of lentiviral infection and the gut microbiome [76–80]. We did find a modest statistically significant positive correlation between TNFα and HIV infection. The TNFα/TNFR pathway has been established as a component of immune activation and the development of viral reservoirs during HIV infection [81]. We found an additional

modest but significant correlation between *P. falciparum* and IL-10. IL-10 is a potent anti-inflammatory cytokine that can ameliorate malaria pathology and promote secretion of antibodies that can protect against malaria reinfection [82, 83].

We also observed significant differences in microbiome composition and subsistence strategy. We had expected that many of the Fulani pastoralists would produce the lactase enzyme given their dairy-rich diets and a reported 50% prevalence of clinical lactose tolerance [84–86]. If the host can successfully take up lactose in the small intestine, then less is passed to the distal colon, which would hypothetically result in lower abundances of gut bacteria capable of catabolizing lactose in lactase-persistent hosts as compared to lactase non-persistent hosts. When we tested for the association of the gut microbiome with the lactase persistence phenotype, we did not detect significant differences in the gut microbes of LP and LNP individuals, including the taxon *Bifidobacterium*, which has been extensively associated with dairy catabolism in populations with majority European ancestry [87]. This result implies that Cameroonians may not possess the same colonic bacteria that catabolize dairy sugar as Europeans.

However, we found that the Fulani pastoralists had higher levels of galactose-metabolizing microbial genes compared to the other Cameroonians. Galactose and glucose are products of catabolism of lactose, and the enrichment of galactose-metabolizing gene pathways may indicate a role of the microbiome in the digestion of dairy products (Additional file 1: Figure S15). Several taxa with small relative abundances cumulatively produced the galactose metabolism pathway result in the Fulani, suggesting that their gut bacteria may act in consortium within an individual to catabolize dairy sugar. This observation is in contrast to a single taxon (i.e., *Bifidobacteria)* being attributed with much of the dairy sugar breakdown in the guts of Europeans. Finally, we observed an enrichment for microbial genes that play a role in lipid metabolism in the Fulani. Unpasteurized cow's milk, such as that consumed by rural pastoralists, contains an average of ∼ 3–4% milk fat (lipids), potentially explaining this observation.

In summary, we present an analysis of 575 Cameroonian subjects, which provides a detailed look at the influence of intestinal parasites and subsistence on the gut microbiome. Further studies would benefit from longitudinally sampling populations and integrating individualized dietary information to distinguish healthy host microbiome structure from parasitized states and to test for the association of microbial diversity with seasonality. Long-term, deeper understanding of the gut microbiome and its interactions with pathogens may provide data useful for optimizing health outcomes in Cameroon and elsewhere.

## Conclusions

This study represents the largest work to date on the correlations between subsistence, polyparasitemia (fecal and blood), and gut microbiota in sub-Saharan Africans. We find high levels of helminths and *Plasmodium falciparum* blood parasites in the Cameroonians and particularly in hunter-gatherers. We establish that co-occurrence of gut parasites is significantly associated with microbial community structure in the gut and that increased diversity is correlated with increasing occurrence of ANTS. We identify putative taxa associated with subsistence type, cytokine response, and the presence of ANTS. We did not find an association between the host's ability to digest dairy and gut

microbiota composition in the Cameroonians. However, Fulani pastoralists possessed an enrichment of lipid and galactose-metabolizing pathways in their gut microbiomes. The gut microbiota are an intriguing potential therapeutic target in the treatment or prevention of helminthiasis, motivating further investigation into the mechanisms behind parasite-host-microbiota interactions.

## Methods

### Participant details

Cameroon samples were collected from nine villages in the Northwest (Ntambang, Sabga), South (Bidou I, Ndtoua), and East (Nkolbikong, Missoume, Njibot, Aviation, Bosquet) regions of Cameroon (Fig. 1a), all of which represented rural communities. The villages in the East and South Administrative regions are located in densely forested areas with primarily tropical monsoon and rainforest climates, while the villages in the Northwest are primarily in tropical savanna climates ((Köppen-Geiger climate classification, https://en.climate-data.org/location/2905/)). The traditional wet season for Cameroon is April through September, and the traditional dry season runs October through March. The Cameroon populations were sampled between January and July. All populations sampled in this study speak languages in the Niger-Kordofanian family. Ethnicity, sample sizes, sampling coordinates, and subsistence classifications are listed in Table S1.

For the Cameroon populations included in this study, we were able to obtain nutrition questionnaires that included types of food grown, foraged or hunted, sold, given to livestock, and consumed, with general frequencies (daily, weekly) of food consumption (Table S2). We also note what sources of water are used, where food is prepared, and if there had been recent periods of food insecurity. Our results were consistent with prior research [4, 9, 50] indicating that Baka and Bagyeli have high-fiber diets, incorporate food from foraging and hunting, and have some small-scale subsistence farming and raising of small livestock (primarily chickens). Bantu agropastoralists rely primarily on small-scale subsistence farming of grains and vegetables for their food and have a larger array of livestock used for meat and trade. The Fulani have similar diets to the Bantu with the notable exception that their diets include a substantial amount of dairy.

DNA extracted from fecal samples from 37 healthy, omnivorous, US participants in the greater Philadelphia area was used here for comparative purposes in the 16S rRNA analyses. Eleven of the US participants self-described their ethnicity as African-American, and 26 self-described as European-American. These samples were collected for a prior research study, the details of which can be found in Wu et al. [23]. Data for age, sex, height, weight, location, and BMI were included in the diversity metric analyses (Table S1).

### Sample collection and storage

Fecal samples were obtained from asymptomatic subjects with no signs of clinical illness and who self-reported as not pregnant. Participants produced a fecal sample in a sterile plastic container that was immediately returned to researchers at the field site. A midsection sample of stool (~ 5 g aliquots) was harvested into a 5 -ml container and immediately frozen in liquid nitrogen. Samples were stored at – 80 °C before transportation to the USA in dry ice, where it was again stored at – 80 °C until extraction.

Relative percentages of lymphocytes, monocytes, eosinophils, basophils, and neutrophils were measured for 570 individuals. For 524 of these individuals, contemporaneous plasma samples were also collected. Blood was drawn into 10 ml capacity BD Vacutainers containing EDTA, and small drops of blood were taken from this tube to measure white blood cell count (HemoCue WBC analyzer and HemoCue WBC cuvettes) and to make thick and thin blood smears on slides for malaria and filarial parasite testing. Following this step, the tube of blood was immediately spun down, and plasma was processed through a Leukolock kit (Ambion Inc.). Plasma was aliquoted into 0.5-ml Eppendorf SafeLock tubes and frozen in liquid nitrogen. The plasma was frozen at − 20 °C, and all samples were analyzed simultaneously.

### Fecal sample processing and DNA sequencing
#### Fecal DNA extraction
Cameroonian and US fecal samples were processed with the same laboratory and computational pipelines for extraction and 16S rRNA analysis. Total DNA from fecal materials was extracted from ∼ 220-mg aliquots using a PSP Spin Stool DNA Plus Kit (Stratec Biomedical; Birkenfeld, Germany) with a modified bead-beating method [88]. PCR and extraction blanks were used to control for reagent and environmental contamination, and all extractions were conducted in a laminar flow hood. Eluted DNA was quantified by fluorometry and stored at − 20 °C.

#### Bacterial 16S rRNA amplicon sequencing
PCR reactions were performed on extracted fecal DNA in triplicate using Accuprime Pfx Supermix (Invitrogen, Carlsbad, CA) and barcoded composite primers with Illumina adapters to amplify the V4 region of the bacterial 16S rDNA genome following the methods of Kozich et al. [89] on a GeneAmp 9700 PCR System. Sequences of DNA primers used in this study are reported in Table S1. PCR conditions were as follows: 95 °C for 2 min, followed by 30 cycles of 95 °C for 20 s, 55 °C for 15 s, 72 °C for 5 min., and then a final elongation step at 72 °C for 10 min. A gene block mock community of eight archaeal species not normally detected in experimental data was used as a positive control following Kim et al. [90] (Table S14). Samples containing the resulting ∼ 250-bp products were pooled, and a subset was visualized by gel electrophoresis on a 1% agarose gel. Library clean-up was performed using SequalPrep Plate Normalization Kits (Invitrogen; Carlsbad, CA), and average library fragment size was checked on a subset of samples using a Tapestation d1000 ScreenTape System (Agilent; Santa Clara, CA). Libraries were quantified using Qubit dsDNA HS Assays (Thermo Fisher Scientific, Waltham, MA) and pooled in equal amounts. Libraries were sequenced on an Illumina MiSeq across 4 runs using 2 × 250 bp cycles in the Bushman Lab. Sequence data are deposited under project accession PRJNA547591 in the NCBI Sequence Read Archive; sample details are in Table S1.

#### Shotgun metagenomic sequencing
A total of 178 fecal DNA sample aliquots were normalized to 0.2 ng/μl DNA, and 1 ng of DNA per sample was used as input for the Nextera XT DNA Sample Prep Kit (Illumina, San Diego, CA) using manufacturer protocols. PCR amplification using unique

combinations of barcoded primers was performed on a GeneAmp 9700 PCR System, and short DNA fragments were removed using AMPure XP bead purification. Library fragment size was visualized on a Tapestation d1000 ScreenTape System (Agilent, Santa Clara, CA), and libraries were quantified using PicoGreen before being pooled in equimolar ratios for sequencing. Three extraction negative controls (denoted "EB" in the metadata) and two library negative controls ("Lib Neg") were included on the run. The pooled library was subjected to a second round of quantification on a BioAnalyzer 2100 (Agilent, Santa Clara, CA), followed by a MiSeq Nano sequencing run for quality control. After this, the pooled library was diluted in hybridization buffer, heat denatured, and paired-end sequenced on an Illumina HiSeq 2500 using V4 reagents in the Penn CHOP Microbiome Core.

### Pathogen testing

#### Microscopy

Stool samples were examined in the field for parasite presence using wet-mount fecal microscopy. Samples were examined with and without iodine staining and visualized with standard light microscopy to identify visible gastrointestinal parasites or parasite ova, including hookworm (species indeterminate with light microscopy), amebiasis (*Entamoeba* spp.), giant roundworm (*Ascaris lumbricoides*), human whipworm (*Trichuris trichiura*), giardia (*Giardia* spp.), human roundworm (*Strongyloides stercoralis*), and flatworms (*Schistosoma mansoni*). We note that there were no positives for *S. mansoni* by light microscopy or by qPCR test of 100 randomly selected samples, and therefore, it was not included in further analyses. Thick and thin blood smear slides were prepared with Giemsa staining to identify blood parasites in the field, including plasmodia (*Plasmodium* spp.) and filaria (*Microfilaria loa loa*, *Microfilaria* spp., *Mansonella perstans*, *Wuchereria bancrofti*). Details on microscopy positivity are in Table S1.

#### Quantitative PCR

DNA oligonucleotide sequences used in qPCRs are listed in Table S15. A gBlocks gene fragment (Integrated DNA Technologies, Coralville, IA) containing parasite target sequences was synthesized and cloned into a TOPO cloning vector, transformed into TOP10 competent *E. coli* cells, and purified with a Qiaprep Spin Miniprep Kit (Qiagen, Hilden, Germany). Purified plasmid DNA was quantified by Picogreen, and the sequence was validated with Sanger sequencing. Plasmids were diluted to a known concentration, and serial 1:5 dilutions were performed to generate a 9-point standard curve. Unknown samples were compared against this standard curve for quantification.

Positive control DNA was extracted from three parasite samples: *Cryptosporidium parvum* from infected mouse stool sample, and *Giardia lamblia* and *Strongyloides stercoralis* from infected canine stool samples using the same methods as human stool DNA extraction. Wells with no template were used as negative controls, and all controls and standards were tested in duplicate. Species-specific primers and probes used in Mejia et al. [24] were used to assay parasite genome copy number for *Ascaris lumbricoides*, *Necator americanus*, *Ancylostoma duodenale*, *Giardia lamblia*, *Entamoeba histolytica*, *Trichuris trichiura*, and *Strongyloides stercoralis* parasites. The pan-*Cryptosporidium* spp. qPCR uses primers and probes from Jothikumar et al., [91], which tests

for ten Cryptosporidium species: *C. hominis*, *C. parvum*, *C. canis*, *C. felis*, *C. parvum-like* (from lemurs), *C. muris*, *C. andersoni*, *C. baileyi*, *C. wrairi*, and *C. serpentis*. All qPCRs were conducted on individual parasites using 384-well MicroAmp EnduraPlate Optical 384-Well Clear Reaction Plates (Applied Biosystems, Waltham, MA) in triplicate on a QuantStudio 6 Flex Real-Time PCR System (Applied Biosystems, Waltham, MA). The total volume per reaction was 7 μl, consisting of 3.5 μl of TaqMan Fast Advanced Master Mix (Applied Biosystems, Waltham, MA), 2 μl of template DNA and 1.44 μl of species-specific primers (final concentration of 900 nM) and probes (final concentration of 250 nM), and 0.06 μl Sigma water (Sigma-Aldrich, St. Louis, MI). qPCRs were run with default parameters and 40 cycles.

In this study, we report the quantification cycle threshold (Ct), which corresponds to the PCR cycle values measuring when fluorescence from template amplification exceeds the background fluorescence. Cycle threshold is an inverse measure of nucleic acid quantity. At least two of the three replicates had to fluoresce within the standard range for the sample to be positive. We had no samples that were positive for *Ancylostoma duodenale*, and thus, this parasite was removed from all downstream analyses. All association testing with the microbiome and other analyses were performed using parasite positivity and parasite genomic copy number taken from qPCR results only.

### HIV testing

Testing for human immunodeficiency virus (HIV) p24 antigen and antibodies to HIV type 1 (HIV-1 groups M and O) and HIV type 2 (HIV-2) was done using a GS HIV Combo Ag/Ab EIA immunoassay (BioRad, Hercules, CA). Testing was performed on 75 μl per sample of thawed plasma according to the manufacturer's instructions. The results were read on a SpectraMax 190 absorbance microplate reader (Molecular Devices, San Jose, CA). In addition to positive controls from the GS HIV Combo Ag/Ab EIA kit, human serum from an anonymous, seropositive donor from the US who had not yet been treated with antiretroviral drugs was used on every test plate.

### Serum cytokine measurements

We measured 21 cytokines from the plasma using a high-sensitivity multiplex cytokine panel (Milliplex MAP Human High Sensitivity T Cell Magnetic Bead Panel, 21-Plex). The cytokine panel included Fractalkine/CX3CL1, granulocyte-macrophage colony-stimulating factor (GM-CSF), interferon-gamma (IFNγ), interleukin (IL) 1β, IL-2, Il-4, IL-5, IL-6, IL-7, Il-8/CXCL8, IL-10, IL-12 (p70), IL-13, IL-17A/CTLA8, IL-21, IL-23, I-TAC/CCL11, MIP-1α/CCL3, MIP-1β/CCL4, MIP-3α/CCL20, and TNFα (tumor necrosis factor α). The panel was run on a Bio-Plex 200 machine using the manufacturer's protocols (Sigma-Aldrich, St. Louis, MI). Cytokine concentrations were determined using standard curves, with the limits of detection for analytes reported in Table S1. A total of 72 Cameroonian samples were analyzed in two batches, with high and low cytokine-specific controls used across both batches. Measurements for each sample and standard curve were performed in duplicate, with the average of the two measurements reported. For two cytokines (MIP-1α/CCL3 and IL-2), we had an insufficient amount of non-NA values to conduct statistical tests, and we removed these from further analysis.

### Lactose tolerance test and lactase persistence phenotype calculation

#### Lactose tolerance test

To test for the association of the gut microbiome with the lactase persistence phenotype, 154 individuals from the Cameroonian cohort were given a lactose tolerance test (LTT) (Ranciaro et al., in preparation). Participants fasted overnight and had baseline glucose measured before beginning the test using either a CodeFree glucometer with SD CodeFree strips or an Accu-Chek Active glucometer with Accu-Chek Active strips. Exclusion criteria included having a baseline glucose outside of 60–100 mg/dl and diabetes. Participants drank a 50-g lactose powder solution (QuinTron, Milwaukee, WI) dissolved in 250 ml water which was equivalent to ~ 1–2 l of cow's milk [51]. Blood glucose was measured in 20-min intervals over the next hour.

#### Lactase persistence phenotype calculation

Glucose values were first adjusted to correct for test strip error using the regression equation $y = 0.985x - 7.5$, where $x$ is the measured glucose value. The maximum rise in glucose level was ascertained by comparing observed glucose values against the baseline value and used to classify the lactase phenotype. Individuals were classified as either lactase persistent (LP) (rise in blood glucose > 1.7 mmol/l), lactase non-persistent (LNP) (rise in blood glucose < 1.1 mmol/l), or lactase intermediate persistent (LIP) (rise in blood glucose between 1.1 and 1.7 mmol/l).

### Anthropometry

#### BMI: height and weight calculations

Height was measured using a Shorrboard Stadiometer (www.shorrproductions.com), with the individual in an erect position with the Frankfurt plane as horizontal as possible. Height was measured with shoes if worn (this was noted in the anthropometry form). Height was recorded in centimeters to the nearest millimeter.

Weight was measured using a set of Seca 876 scales and recorded in kilograms to one decimal place. Weight was measured with shoes if worn. Care was taken to ensure that the scales were firmly seated and level.

#### Temperature

Temperature was measured in triplicate on a non-contact infrared thermometer.

### Quantification and statistical analysis

#### ASV inference

The V4 region of the bacterial 16S rRNA gene was sequenced on the Illumina MiSeq platform across 4 MiSeq runs. FASTQ files were generated from raw BCL files using "configureBclToFastq.pl" (Illumina, San Diego, CA), and paired-ends were assembled using the QIIME2 pipeline. All sequences went through quality filtering, demultiplexing, chimera removal, denoising, and merging using the demux and DADA2 plugins with default settings. DADA2 produces an amplicon sequencing variant (ASV) table that can resolve unique sequences down to single-nucleotide differences and attaches biological meaning to sequences independent of a reference database. All ASV feature tables were then merged (https://github.com/marubel/R-ubelMisc). We used a classifier

that was pretrained on the V4 region targeted by the 515F and 806R primer sets [92, 93] with 99% OTU sequence similarity using the most recent version of the Green-Genes (http://greengenes.secondgenome.com/) database. Sequences classified as mitochondria and chloroplasts were removed. To conduct phylogenetic analyses of microbiome sequences, sequences were aligned with MAFFT [94] and a phylogenetic tree was produced with FastTree2 [95] using default settings. Sequences derived from plastids and mitochondria were removed. A gene block mock community of eight archaeal species not normally detected in experimental data was used as a positive control across runs (Table S14), following the methods used in Kim et al. [90]. The four gene block controls, 12 negative extraction, and PCR controls were dropped from further analysis. This produced a total of 14,138 ASVs.

### Shotgun metagenomic sequencing processing and analysis

Illumina BaseSpace output metrics from the shotgun metagenomics run are available in Tables S16-S18. In brief, there were 1,820,262,487 raw reads and 1,820,262,480 reads that passed the Illumina chastity filter. The Illumina chastity filter measures the ratio of the intensity base call divided by the sum of the brightest and second brightest intensity base calls. Raw shotgun metagenomic data files were de-multiplexed and converted from BCL to FASTQ using bcl2fastq (Illumina, San Diego, CA), which drops unassigned reads, including those mapping to PhiX, a common sequencing control. Demultiplexed FASTQ files were analyzed using the Sunbeam pipeline [28], as detailed in https://github.com/sunbeam-labs/sunbeam/. In short, quality filtering was done using default settings of Trimmomatic [96] (reads below 36 bases, trailing or leading bases with quality scores below three, and base reads scanned in a 4-nt sliding window with average quality/base < 15 were dropped), and adapters were trimmed from sequences with Cutadapt (fwd_adapters ["GTTTCCCAGTCAC-GATC," "GTTTCCCAGTCACGATCNNNNNNNNNGTTTCCCAGTCACGATC"] and rev_adapters ["GTTTCCCAGTCACGATC," "GTTTCCCAGTCAC-GATCNNNNNNNNNGTTTCCCAGTCACGATC"]) software [97]. This effectively dropped the two library negatives and the three extraction blanks from further shotgun analyses. FastQC (Babraham Bioinformatics) was used to assess the read quality on read pairs surviving quality filtering (Table S17). Low complexity sequences were masked using https://github.com/eclarke/komplexity with a normalized complexity score of < 0.55. For $k = 4$, this scores that across a 64–120-bp region, the sequence is strongly suggestive of being low-complexity, repetitive sequence, and thus is unlikely to be informative. Reads that mapped to a human reference sequence (Genome Reference Consortium Human Build 38, GRCh38) were identified using bwa [98]. Sample reads with > 60% of the read fraction mapping to GRCh38 or with a percent identity > 50% were removed. Per sample, non-host (microbial) reads can be found in Table S18. The output from the Sunbeam quality control was inspected manually using the sbx_report extension (https://github.com/sunbeam-labs/sbx_report). This produced a total amount of 1.65 billion host-filtered, quality-controlled reads (controls not included). This amounts to a median of 8.5 million reads and an average of 9.3

million reads per sample. KrakenUniq [30] was used to classify human-filtered, quality-controlled reads using the Sunbeam extension sbx_kraken_uniq (https://github.com/ArwaAbbas/sbx_kraken_uniq) on a low-complexity masked database of bacterial, archaeal, viral, fungal, and protozoal sequences from NCBI nt (downloaded 13 December 2018). Classifications reported at the genus and species level are reported as relative abundances and were further filtered based on meeting a threshold of the number of reads and read to *k*-mer ratio, as described in the figure legends. Shotgun metagenomic reads were also classified using two alternate methods for comparison: MetaPhlAn2 [29] on the MetaPhlAn2 mpa_v20 database using the sbx_metaphlan Sunbeam extension (https://github.com/sunbeam-labs/sbx_metaphlan/) and the GreenGenes 16S database using Kraken2 (https://ccb.jhu.edu/software/kraken2/). As expected, given the differences in the database used, and known 16S primer biases, the relative proportions of bacterial genera classified by amplicon and shotgun sequencing did not always correspond. In a comparison across MetaPhlAn2, KrakenUniq, and 16S Greengenes classification for both V4 and shotgun data, comparisons between V4 and shotgun data annotated against the same Greengenes Database most closely paralleled each other (Additional file 1: Figure S8). Second to that, the KrakenUniq database showed the least divergence between V4 and shotgun taxonomic identification. Some relevant genera were highly divergent in KrakenUniq to V4 comparison, including *Klebsiella* and *Eubacterium*, which were absent from V4 datasets. This could be the result of primer or database bias or nomenclature differences.

We noted that average 16S copy number across *Prevotella* and *Bacteroides* genomes varies (average 16S copies across 24 species of *Prevotella* = 4, average across 24 species *Bacteroides* = 5.3) (https://rrndb.umms.med.umich.edu/genomes/). In our shotgun metagenomics data, the average size of *Bacteroides* genomes was 5.3 Mbp and the average size of *Prevotella* genomes was 2.7 Mbp. Larger genome size and more 16S copy numbers in *Bacteroides* could account for some of the variations we see in higher relative abundances of this taxon in 16S versus shotgun sequencing compared to *Prevotella*. Quality-controlled reads were aligned to the KEGG database [41, 42, 99] using DIAMOND [43] with an *e* value cutoff $1 \times 10^{-6}$. The resulting KO numbers were mapped to the associated pathway, module, and enzyme identifiers (https://github.com/marubel/kegg-r-ator). Where a single KO mapped to multiple pathways, enzymes, or modules, weighted counts were used such that each KO contributed a single count equally distributed across all pathways, enzymes, or modules mapped to it.

As a further validation, we compared the percentages of reads that could be classified to the family level for shotgun metagenomic data KrakenUniq with metagenome-assembled genome (MAG) taxonomic information. We obtained the sequences for the representative MAGs from http://segatalab.cibio.unitn.it/data/Pasolli_et_al.html [33]. Using the taxonomic information provided in the published metadata and corresponding NCBI Taxonomy identifiers [100] kindly provided by the members of the Segata lab, we constructed a custom Kraken2 database [101] and classified the quality-controlled metagenomic reads described above using Kraken2 as run by version 3.0 of the Sunbeam pipeline [102]. Downstream visualization was performed using R and packages in the tidyverse, including ggplot2 [103–105].

To detect helminths in shotgun metagenomic sequences, human-filtered, quality-controlled reads were aligned to nine representative genomes downloaded from NCBI Genome (*Trichuris trichiura, Ancylostoma duodenale, Strongyloides stercoralis, Ascaris lumbricoides, Necator americanus, Entamoeba histolytica, Giardia lamblia, Cryptosporidium hominis*, and *Cryptosporidium parvum*) or WormBase ParaSite (*Ascaris lumbricoides*). Read alignments were performed using hisss (https://github.com/louiejtaylor/hisss) [99] with the following modifications to Bowtie2 (-end-to-end --very fast).

The results from all analyses were visualized in R [106] using packages tidyverse [107], taxonomizr [108], magrittr [109], reshape2 [110], ggplot2 [104], vegan [111], and ape [112]. Pathway, module, and enzyme differential enrichment were calculated using LefSe [44], which produces absolute values of log10-transformed LDA scores as effect sizes for a given taxa/group. The code used to generate LefSe metrics and heatmaps can be found at https://github.com/ressy/LEfSe. FDR correction was applied to all LefSe results.

### Diversity metric analysis

For 16S data, alpha diversity was assessed by three metrics in QIIME2: the observed number of OTUs (bacterial "richness"), Shannon's Index (bacterial abundance and evenness of species present), and Faith's Phylogenetic Diversity Index [113], which incorporates phylogenetic relatedness of taxa in each sample. Beta diversity was assessed using the Bray-Curtis dissimilarity index, which measures abundance information, and the Jaccard similarity coefficient, which measures presence/absence information. Both metrics quantify the compositional dissimilarity between two different samples, bound between 0 and 1, where 0 is the same composition and 1 is maximally dissimilar composition. Metadata covariates were tested for associations with the microbiome using permutational multivariate analysis of variance (PERMANOVA) tests in R using the "adonis" function of the vegan package. PERMANOVA tests were done on both unweighted and weighted UniFrac distance matrices, which allows for comparison of intragroup and intergroup distances using a permutation scheme to obtain $p$ values. PERMANOVA tests were done with 999 randomizations. Low variance ASVs were removed for differential sample abundance analysis, which was determined with the edgeR [114] and phyloseq [115] packages in R. False discovery rate correction was performed on all resulting PERMANOVA and differential abundance $p$ values using the Benjamini-Hochberg (FDR/BH) criterion.

For shotgun metagenomic data, alpha diversity was calculated with Simpson's and Shannon's Diversity Indices. Simpson's Diversity Index is a measure which considers the number of species present, as well as the relative abundance of each species. The distribution of reads classified at the prokaryotic genus level and at > 1% abundance in each fecal sample ranged between a minimum of 195,404 and a maximum of 7,662,130 reads. For diversity metrics, reads were randomly subsampled to 150,000 reads. The R function vegdist in the vegan package [116] computed dissimilarity indices using Bray-Curtis, which quantifies the compositional dissimilarity between two different samples.

### Random forests

Random forest classifiers (RFCs) were implemented using the randomForest package [117] in R. Parameters included 5001 decision trees, which were trained on taxa abundance data consisting of 14,138 ASVs for our 16S dataset and 20,844 taxa for our

shotgun metagenomics dataset. Binary variables (e.g., positive, negative) were analyzed using classificatory RFC, and continuous variables (e.g., cytokine values) were analyzed using regression RFC. Discriminating taxa were identified by random forest using importance values, which were calculated as mean decrease in Gini index for classification random forests and percent increase in mean squared error (%IncMSE) for regression random forests. The top ten importance values are reported for each random forest test. The error of the model was assessed using out of bag (OOB) error. To increase the classifier's ability to detect true positives, we introduced a positive control consisting of statistical noise with a probability density equal to the values within the variable of interest (i.e., values within those present in the classification/regression variable) (https://github.com/marubel/R-ubelMisc). In RFC classification, the prediction accuracy of the model was tested by randomly sampling half the samples of whichever of the two groups was smaller and training the classifier on this subset, for variables that had a minimum of at least five samples in each group. For example, if there were 80 positives and 100 negatives for a parasite, 40 positive and 40 negatives would be input as the training set for the classifier. RFC uses bootstrap sampling, which means that some training set samples in each downsampled category will be selected more than once, over a total of 5001 iterations, to produce a consensus tree.

### Co-occurrence analysis

Probabilistic co-occurrence analysis was done in R using the cooccur package [118, 119]. Parasite pairs were removed if they shared less than one site. For parasite groups with presence/absence data, all pairwise combinations were tested using the hypergeometric distribution, which produced an observed-expected ratio and effect size for significant species combinations. All $p$ values for FDR were corrected for multiple test correction. Sample pairs were dubbed "random" if they did not significantly differ from their expected number of co-occurrences and if they did not deviate by < 10% of the total number of sites, following the power analysis recommendations in Veech et al. (2013) [119]. Pairwise combinations were visualized as heatmaps using ggplot2, where parasite combinations are measured from most negative to most positive interactions (left to right in the heatmap). Deviation from expected co-occurrence values was plotted against observed values.

### Correlation analysis

Correlation analysis was conducted using the corrplot package in R [25]. Correlations were calculated using Spearman's non-parametric rank-based correlation tests to control for potential outliers and hierarchical clustering was used to aggregate the correlation matrix. Correlation values and figures are available in Tables S4, S10, S11, and S12 and Sup. Figs. 3 and 13. Correlation values were considered significant if $p$ values were less than or equal to 0.01. Cytokine plots incorporated 19 cytokines across 72 Cameroonians. For the metadata correlation analysis, we excluded 82 samples due to null values in metadata variables ($n = 492$ Cameroonians). Variables for the metadata correlation analysis were as follows: ANTS binary, total parasites, total parasites and blood parasites, total ANTS, body mass index, average temperature, WBC, subsistence, population, region, HIV status, neutrophil, lymphocyte, monocyte, eosinophil, *Wuchereria bancroftii, Mf. M. perstans, P. falciparum, Mf. Loa loa, W. bancrofti, Microfilaria* spp., and highly positive ANTS.

## Supplementary information

---

**Additional file 1.** All supplementary figures.

**Additional file 2.** All supplementary tables.

**Additional file 3.** Review history.

---

### Acknowledgements

Many thanks to Boris Striepen, Robert Greenberg, Richard Marcantuno, Thomas J. Nolan, James Lok, and Ronald Collman for generously sharing parasite and pathogen samples for use as experimental controls. Dan Beiting, members of the Penn CHoP microbiome core, the Penn Human Immunology Core, and the Bushman and Tishkoff labs gave invaluable project help and suggestions. We want to recognize Young Hwang (University of Pennsylvania), Elizabeth Loy (University of Pennsylvania), Alexa Avitto (University of Pennsylvania), and Jaanki Dave (The Geisinger Commonwealth Medical College) for the assistance with extractions and assays. We thank Moreno Zolfo and Fabio Cumbo of the Segata Lab for sharing the taxonomic classifications of representative MAGs. We thank Peter Kfu, Eric Mbunwe, and Grace N. Tenjei for the fieldwork in Cameroon; William Beggs and Lillian Chau for the sample coordination; the Cameroonian villagers for their willingness to participate in this research; and the Cameroonian Ministry of Public Health and the Cameroon National Ethics Committee for permission to collect samples in Cameroon. We also thank the community outreach groups and personnel, including the Mbororo Social and Cultural Association (MBOSCUDA), Mr. Sali Django, the Association Culturel pour le Dévelopment Bagyeli/Bakola de l'Ocean (BACUDA), and the Centre d'Action pour le Dévelopment Durable des Autochtones Pygmées (CADDAP) for their support, as well as the partnership with the Faculty of Medicine and Biomedical Sciences of the University of Yaoundé 1, Cameroon.

### Peer review information

### Review history

The review history is available as Additional file 3.

### Authors' contributions

The study design was by SAT, FDB, and MAR. Cameroon biological samples and measurements were collected by MAR, VNN, JYF, EN, and AE, with the coordination of SAT, CF, and AKN. US fecal DNA (COMBO) were analyzed in collaboration with FDB. Fecal DNA extraction, 16S rRNA, shotgun sequencing, qPCRs, and HIV testing were conducted by MAR with support from FDB and SAT. MAR, LJT, AA, and AC performed the data analyses, with assistance from CT and KB. The manuscript was written by MAR with assistance from LJT and AA and was edited by SAT in consultation with FDB. All authors have read and approved the manuscript.

### Funding

This research was supported in part by the Lewis and Clark Fund, the University of Pennsylvania, the Leakey Foundation, the Wenner-Gren Foundation (9299), an NIH training grant in Parasitology (5T32AI007532-18), and the National Science Foundation (BCS-1540432) to MAR and to SAT (BCS-1317217). SAT also provided support from an American Diabetes Association Pathway to Stop Diabetes grant (1-19-VSN-02). Additional support came from NIH awards to SAT (1R01DK104339-01, 1R01GM113657-01, and R35 GM134957-01) and to FB (R01-HL113252, R61-HL137063, U01-HL098957, R01-HL087115, K24-HL115354). FB also contributed assistance from the Penn Center for AIDS Research (P30-AI045008) and the PennCHoP Microbiome Program.

### Availability of data and materials

16S amplicon and shotgun metagenomic sequence data are deposited under project accession PRJNA547591 in the NCBI Sequence Read Archive (SRA) [120]. Human Microbiome Project samples used for comparative analyses in shotgun data can be found in the SRA under the following accession numbers: SRR1804648, SRR1565914, SRR1803892, SRR1803862, SRR1804618, SRR1803903, SRR1803864, SRR1804203, SRR1803877, SRR1804107, SRR1804009, SRR1804055, SRR1804676, SRR532163, SRR1804148, SRR1804756, SRR1031154, SRR1804119, SRR1803355, SRR1803358, SRR1804539, SRR1564387, SRR512768, SRR1803287, SRR1031102, SRR1804688, and SRR1804086.

### Ethics approval and consent to participate

Written, informed consent was obtained from all study participants, and research/ethics approval was obtained from the following institutions prior to the start of sample collection: Institutional Review Board of the University of Pennsylvania (IRB # 807981), the Cameroonian National Ethics Committee and the Cameroonian Ministry of Public Health. All subjects provided written informed consent for the collection and analysis of samples. All samples were coded with an alphanumeric identifier to protect participant confidentiality.
All experimental methods were in accordance with the Helsinki Declaration.

### Consent for publication

All participants provided consent for publication of study results of the collected biomaterials paired with anonymized metadata information.

### Competing interests

The authors declare that they have no competing interests.

 

## Author details
[1]Department of Anthropology, University of Pennsylvania, Philadelphia, PA 19104, USA. [2]Department of Genetics, Perelman School of Medicine, University of Pennsylvania, Philadelphia, PA 19104, USA. [3]Present Address: Department of Radiology, Center for Translational Imaging and Precision Medicine, UC San Diego, San Diego, CA, USA. [4]Department of Microbiology, Perelman School of Medicine, University of Pennsylvania, Philadelphia, PA 19104, USA. [5]Present Address: Department of Pathology and Laboratory Medicine, Children's Hospital of Philadelphia, Philadelphia, PA 19104, USA. [6]Division of Gastroenterology, Hepatology, and Nutrition, The Children's Hospital of Philadelphia, Philadelphia, PA 19104, USA. [7]Johns Hopkins Cameroon Program, Yaoundé, Cameroon. [8]Department of Microbiology, Hematology, Parasitology and Infectious Diseases, Faculty of Medicine and Biomedical Sciences, University of Yaoundé I, Yaoundé, Cameroon. [9]Department of Neurology, Faculty of Medicine and Biomedical Sciences, Yaoundé Central Hospital, Yaoundé, Cameroon. [10]Department of Anthropology, Faculty of Arts, Letters and Social Sciences, University of Yaoundé I, PO Box 755, Yaoundé, Cameroon. [11]Mbalmayo District Hospital, Mbalmayo, Cameroon. [12]Department of Pharmacotoxicology and Pharmacokinetics, Faculty of Medicine and Biomedical Sciences, University of Yaoundé I, Yaoundé, Cameroon. [13]Department of Neurology, Central Hospital Yaoundé, Yaoundé, Cameroon. [14]Neuroscience Lab, Faculty of Medicine and Biomedical Sciences, University of Yaoundé I, Yaoundé, Cameroon. [15]Brain Research Africa Initiative (BRAIN), Yaoundé, Cameroon. [16]Department of Biology, University of Pennsylvania, Philadelphia, PA 19104, USA.

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

## 
