## [**Additional file 3.** Review history. · Genome Biology]

Review History

First round of review

Reviewer 1

Comments to author:

Summary:

In this manuscript, the Rubel et al performed 16S rRNA gene and metagenomic sequencing on stool samples from 575 Western Africans in Cameroon. They studied individuals who practice a variety of lifestyles (agropastoralist, hunter-gatherer) and compare these microbiomes to those of individuals from the US (Philadelphia). They find differences in microbiome composition that correlate with subsistence practices/lifestyle practices and self-reported ethnicity (they do not use host genetic features). They present a large number of descriptive analyses, including an analysis of lactose consumption and microbiome composition as well as an analysis of eukaryotic protists and parasites in these samples. They go on to finally look at the relationship between a subset of eukaryotic parasites and a panel of cytokines.

Impressions: The work presented is rigorous, and the populations studied were of great interest. Additionally, the authors make excellent use of positive and negative controls throughout. However, the manuscript lacks a strong take-home message, and at times is meandering. Despite the fact that I find the subject matter the investigators chose to study very interesting, I had a hard time getting excited about the many results presented. Many correlations are tested (blood vs. gut parasites, gut parasites vs. gut bacteria, gut parasites and bacteria vs. cytokines, gut parasites/bacteria vs. lactase persistence) and classifiers are generated - but none of this amounts to much in terms of a clear model of how these organisms are interacting. I am left wondering what the next step should be in this research direction and think that the manuscript would be more interesting if the investigators went deeper into any angle of the manuscript as opposed to staying broad in their analysis of the metagenomic data that were generated.

Major comments:

#. The abstract states, "African populations provide a unique opportunity to interrogate host-microbe coevolution and its impact on adaptive phenotypes, thanks in part to their genomic, phenotypic, and cultural diversity." - yet, there is no data provided on the genotypic and phenotypic diversity within the populations studied here. Given the framing of the abstract, I was surprised to see this excluded from the analyses presented. Certainly, the investigators have reported in individuals who have varied cultural dietary experiences, but the current manuscript only offers limited information on these varied practices.

#. The results section is somewhat meandering and it is difficult to draw a strong narrative thread through it. This could be aided by section headers or a stronger organizational structure.

#. What does this manuscript offer above and beyond the work published by Gomez et al (Cell Report 2016)? How concordant/discordant are the results? I appreciate the use of metagenomic approaches to characterize the eukaryotic parasite component of the stool metagenome - but beyond this, what else do we learn that is new? Also, previous work suggested that Bantu populations are enriched for Firmicutes compared to BaAka - unless I am mistaken, this is not seen in the current study.

#. I'm curious about the classifiability of the authors' newly generated stool metagenomic read datasets using the recently described MAG databases.

#. I recommend that the authors be careful not to conflate correlation with causation - for example, "our sample set also allows us to ask how the different baseline microbiotas in rural populations are impacted

by colonization with multiple parasites". Another example, " Infection with ANTS parasites, in turn, was associated with elevated levels of TH1, TH2, and proinflammatory cytokines, indicating engagement of multiple immune mechanisms." And yet another example, "These data document transkingdom interactions" - I'm not sure that "interactions" are actually documented, per se (though I acknowledge that the last example is a minor point).

#. One of the most interesting findings in this manuscript, to me, is the finding that Bacteroidales is associated with higher proportion of ANTS infection. It would be helpful if the authors speculated about the potential reason for this finding. This is particularly so as Bacteroidales appears to be negatively correlated with *E. histolytica* in humans and in murine models, helminths are associated with a reduction of Bacteroidales.

#. One major missing piece of metadata is diet - do the authors have any data on this important factor?

#. Detection of parasites using microscopy is notoriously insensitive - what was the training of the individuals who evaluated the samples? Classically, ova and parasite tests are done on sequential stool samples to improve the sensitivity of the test.

#. The inclusion of children (minimum age 4 in some categories) with adults would seem to be a substantial confounder. This is especially so in considering the lactase persistence analysis. Is there a reason that the authors favored combining children and adults into the same analysis? This is rather unusual.

#. For the kraken database build, I worry a bit that the NCBI nt dataset includes many poorly labeled, non-curated genomic sequences, which may introduce false positive findings into the classifications. Is there a reason that this was selected as the choice of input data for the database build? A few words on this choice would be helpful to the reader. Also, as noted above, why didn't the authors use the recently published MAG datasets? In particular, the Pasolli et al dataset (Cell 2018) would seem to be a good fit given datasets from Ethiopia and Madagascar in that MAG build.

#. Schistosomiasis is a noted issue in Cameroon, and others (Ratard et al, PMID 2115306) have described a high rate of intestinal carriage of Schisto in Cameroon - is there a reason this was not detected in the studied cohort? Was it missing from the databases used for classification?

Minor comments:

#. What is a "longevity regulating pathway"? This needs some explaining/context.

#. Perhaps a small or semantic point - but what separates a parasitic "infection" from "colonization". Given the high rate of colonization that was observed, is it possible that all colonization with candidate disease-causing parasites are not true "infections"? I realize this may be beyond the scope of the work presented, but given how interesting the parasite findings are, I think it is worthy of discussion and potentially having a parasitologist on the team comment on this.

#. While interesting, the lactose nonpersistence vs. persistence section came across as a non sequitur

#. The speculation that "this higher diversity associated with helminth infection could result from parasite-induced rapid peristalsis..." is intriguing, but would be stronger and more convincing if the association between parasites and peristalsis/epithelial shedding was cited.

#. Re Entamoeba co-infection - are the authors sure that the findings of multiple taxa of Entamoeba being

present are the result of co-infections as opposed to infection with a single intermediate organism for which there isn't yet a reference genome? What is the evidence for co-infection vs. this alternative model? I wonder whether co-detection of multiple species is a legitimate finding or if it is the result of "taxon splitting" by Kraken. One could use bracken to re-estimate species-level abundance (per the kraken documentation, it appears that bracken is compatible with Kraken-Uniq).

#. While blood parasitemia (malaria) was investigated in this manuscript, it is not featured in most of the analyses that are presented, which makes it seem like a disconnected aspect of the manuscript.

#. The authors suggest that longitudinal data would be helpful as a follow on, why? Is there a strong seasonality to diet and exposures in the Cameroonian individuals studied in this project? In comparison, the Hadza have a very strongly seasonal diet and lifestyle.

#. Please include details of the primers used for 16S sequencing.

#. What is "Sup.Table: Metadata"?

#. Presumably exclusion criteria for the baseline glucose was EITHER glc outside of 60-100 OR a diagnosis of Diabetes? Please clarify the writing of this sentence.

#. Perhaps I missed this, but how are the authors inferring genome size of Prevotella and Bacteroides using their shotgun data? I didn't see a section on genome assembly and binning, which presumably was used for this purpose? Also, assembly of 16S into the metagenome assembled genomes is notoriously difficult - thus, I'm not sure I'd make much of the finding around the "average 16S copies" in Prevotella vs. Bacteroides.

#. The Blastocystis paragraph in the results comes out of left field, and doesn't really flow with the narrative. Perhaps it is somewhat misplaced?

#. Typo:

- "differentially enriched amount the three subsistence groups"

Reviewer 2

Comments to author:

In this study by Rubel et al., the relationship between parasite load, the microbiome, and population/subsistence strategy is explored. An impressive collection of >500 samples was collected across Cameroon, including fecal samples for microbiome and parasite characterization and blood samples for immune parameter characterization. There are an impressive amount of analyses relating all of these data types together, showing that microbiome diversity tends to increase with the number of parasite infections in the host; microbiomes differ between population/subsistence strategy; there are relationships between immune cell fractions, temperature, and parasite count; particular microbes and microbial functions are associated with parasites, and there are differences in microbial gene content depending on lactase persistence status. The authors do a careful, diligent job, clearly describing methodology in detail, using appropriate statistics, and performing sub-analyses where there's concern of confounding between populations. Overall, this is a novel, carefully done study that gives a large number of insights into the relationship between the microbiome, parasites, and host physiology in industrializing populations, which is of growing interest to the microbiome field.

I have just a couple of very (very) minor comments:

- One of the more surprising findings of the study was the fact that lactase persistence wasn't associated with microbiome composition in the Cameroonians, which is different from what is observed in European populations. It may be beneficial to include a few more details in the results section about these investigations. As written, it sounds like only overall composition was examined, and not associations between LP and specific microbes. Was the association between LP and Bifidobacteria tested (specifically, since that has what has been observed in European populations)? If an association wasn't observed (which I imagine it wasn't, or it would be included here), is the abundance of Bifidobacteria lower in these populations as compared to European populations? Additionally, the association between LP and Bifidobacteria is only apparent in European populations when there's milk in the diet (<https://www.ncbi.nlm.nih.gov/pubmed/27694959>). Did dairy consumption recorded for the different populations of this study? Is there potentially only an association in those individuals who are LP in the pastoralist population? Finally, given that there is a difference in galactose and lipid metabolism pathways in the pastoralists (which is a fascinating result), can those genes be traced back to particular microbes using the metagenomic data (for example, do those genes increase in abundance because some group of microbes other than the Bifidobacteria increase in abundance in those individuals)?
- Ensure that the S of 16S is consistently capitalized throughout the manuscript.
- Legend of Figure 3F: Fill in the correct Supplemental Figure number (currently XX)

We thank the reviewers for their helpful comments and appreciate the opportunity to provide answers to their questions and to revise the manuscript accordingly, which we believe improves the paper. Below are specific responses to reviewer questions, binned by their subject matter where applicable.

Reviewer 1, Major Comment #1: The abstract states, "African populations provide a unique opportunity to interrogate host-microbe coevolution and its impact on adaptive phenotypes, thanks in part to their genomic, phenotypic, and cultural diversity." - yet, there is no data provided on the genotypic and phenotypic diversity within the populations studied here. Given the framing of the abstract, I was surprised to see this excluded from the analyses presented. Certainly, the investigators have reported in individuals who have varied cultural dietary experiences, but the current manuscript only offers limited information on these varied practices.

Author Response: We note that this sentence is meant to refer broadly on the informativity of African populations in microbiome research and, as such, it is correct. The focus of the research presented here is on the gut microbiome and associations with parasites, though we do also examine correlations with subsistence and phenotypes such as BMI and lactase persistence. In the revised draft, we include dietary information in Sup. Table 2 and show that these dietary results broadly corroborate those from prior studies with similar and/or overlapping populations (Please see our response to *Reviewer 1, Major Comment #7*). We have provided additional citations for studies focused on Cameroonian genotypic and phenotypic diversity and have added these into background, provided below (full references are at the end of this document):

“Here we focus on populations from Cameroon that are genetically, linguistically, phenotypically, and culturally diverse [1–5]”

Reviewer 1, Major Comment #2: The results section is somewhat meandering and it is difficult to draw a strong narrative thread through it. This could be aided by section headers or a stronger organizational structure.

Author Response: In response to the reviewer’s point, we have added introduction and further description to several sections (See response to Reviewer 1, Minor Comment 3) and we now provide a paragraph introducing the sections of the results:

“In the section below we first present the populations studied, then quantification of infections potentially influencing composition of the gut microbiota. We then compare associations of parasitemia and gut microbiome structure, first using 16S rRNA gene tag sequencing, then using shotgun metagenomics. Lastly, we analyze a few additional features of lifestyle-microbiome interactions in the Cameroonian populations studied.”

We added the following subheadings to the results section:

“Association of microbiome structure and blood-borne pathogens
“Analysis of additional eukaryotic gut organisms”
“Features of subsistence groups in metagenomic data”
“Association of ANTS parasite detection and bacterial gene content”
“Comparison of microbial data to human immune responses reported by cytokine production”
“Further analysis of gut microbiota, subsistence strategy, and diet: gut microbiota are not associated with the lactase persistence phenotype in Cameroonians”

Reviewer 1, Major Comment #3: What does this manuscript offer above and beyond the work published by Gomez et al (Cell Report 2016)? How concordant/discordant are the results? I appreciate the use of metagenomic approaches to characterize the eukaryotic parasite component of the stool metagenome - but beyond this, what else do we learn that is new? Also, previous work suggested that Bantu populations are enriched for Firmicutes compared to BaAka - unless I am mistaken, this is not seen in the current study

Author Response: We appreciate the opportunity to clarify what this research offers beyond Gomez et al. (2016) and describe what we learn that is new. To our knowledge, this is the largest single study to date of nonindustrialized microbiomes (N=575). As compared to Gomez et al. (2016) [6] and Morton et al. (2015) [7], we move beyond amplicon sequencing to also include shotgun metagenomics and test for association of the gut microbiota with pathogen single and co-infections across blood and fecal tissues. We include functional measurements of physiology including lactose tolerance tests and cytokine data, and test for association of these with gut microbiome sequence data. We include hunter-gatherers, agropastoralists, and Fulani pastoralists from Cameroon, the latter of which have never had their gut microbiomes described before. Gomez et al. (2016) did not include reporting of pathogens and Morton et al. (2015) diagnosed pathogens by light microscopy only, whereas we also used qPCR for more sensitive and specific detection. We then used this data to consider cooccurring infections in addition to individual infections when testing for association with the gut microbiome, as this more accurately reflects the *in vivo* state of the host. We find that gut microbiome composition can be used to predict infection with a significantly cooccurring group of helminths with higher accuracy than any other metrics except Country and Subsistence, indicating the importance of including pathogen variables (and in particular, helminth infection) when considering correlates with microbiome composition. We added the following comment into the Discussion to emphasize this point:

“Microbiome composition could be used to predict ANTS parasite coinfection with greater accuracy than all other tested variables save country and subsistence, indicating the importance of considering gut parasite coinfections when studying microbiome composition in populations known to have helminthiasis.”

We identify particular taxa that are highly correlated with ANTS helminth infection using different metrics, and find that these taxa are also associated with increased IL5 production. Like

prior research, we find that subsistence shapes gut microbiome composition but uniquely, identify a pathway (Galactose catabolism) related to the digestion of dairy sugar enriched in the pastoralist population, and thus hone in on putative dairy catabolism as a relatively unexplored area in the study of rural African gut microbiota. However, we identify for the first time a galactose catabolism pathway that is highest in the pastoralists, and does not come from *Bifidobacteria*, which is often seen in the gut microbiomes of individuals capable of breaking down dairy in Europe, indicating that other bacteria may act in consortium to breakdown milk sugar in the host colon.

To reviewer 1's final question in Major Comment #3, Reviewer 1 is correct that we do not see an enrichment of *Firmicutes* in the Bantu as compared to the Baka. From 16S rRNA sequencing, we find that the median proportional abundance of *Firmicutes: Bacteroidetes* in the Baka was approximately 1:1 (42.52% *Firmicutes* to 37.86 *Bacteroidetes*), and for the Bantu it is also approximately 1:1 (39.3% *Firmicutes* to 40.27% *Bacteroidetes*). Although Gomez et al. sampled Baka individuals (called "BaAka" in their study), and we include Baka individuals in ours, we do not anticipate that our results will highly overlap, for several reasons: 1) The Baka that Gomez et al. sampled from the Dzanga Sangha protected areas of CAR are >350 km (217 miles) from our closest Cameroonian sampling location, and while sharing an ethnic identity and ancestry, we don't expect that the Baka and the Bantu in CAR and Cameroon will have identical diets; 2) Gomez et al. sampled a much smaller number of individuals than we did (28 Baka and Bantu, as compared to our 117 Baka and 322 Bantu). Our study draws from a larger, and thus arguably more representative, segment of these populations. 3) Importantly, Gomez uses V1/V3 amplicons in their research, and Morton et. al. use V5/V6, whereas we used V4. Primer and amplicon choice in sequencing introduce bias in the 16S rRNA amplicon metagenomics and this can impact assessment of microbial abundances [8,9]. Primer bias is a known caveat in the field and consequently, sequencing data between different 16S rRNA amplicons cannot be robustly compared.

Reviewer 1, Major Comment #4: I'm curious about the classifiability of the authors' newly generated stool metagenomic read datasets using the recently described MAG databases.

Reviewer 1, Major Comment #10: #. For the kraken database build, I worry a bit that the NCBI nt dataset includes many poorly labeled, non-curated genomic sequences, which may introduce false positive findings into the classifications. Is there a reason that this was selected as the choice of input data for the database build? A few words on this choice would be helpful to the reader. Also, as noted above, why didn't the authors use the recently published MAG datasets? In particular, the Pasolli et al dataset (Cell 2018) would seem to be a good fit given datasets from Ethiopia and Madagascar in that MAG build.

Author Response: We had most of our analyses completed using the KrakenUniq database by the time the Pasolli et al. 2019 MAGS paper was published. We acknowledge that the nt database has some poorly labeled and curated genome sequences, however, its large size allowed us to interrogate organisms that do not have deposited reference genomes, and this is very relevant for

the human populations studied here. In order to minimize false-positive annotations using such a database, we used KrakenUniq, which calculates unique k-mers as a proxy for better coverage of an authentically detected organism. We ensured a minimum number of unique reads to an organism for reporting as well as another metric (Total Reads/Total K-mers) to approximate the confidence value of that assignment.

Following the suggestion of Reviewer 1, we performed a comparison of read percentages aligned to the family level for the shotgun metagenomic data using both KrakenUniq and the MAGs database (new Sup. Fig. 9) [10]. We used a fast, memory-intensive approach—read taxonomic classification with kraken2. We engineered a custom taxonomy map using the taxonomic classification of the representative MAGs [11] and the NCBI taxonomy IDs (taxids) kindly provided by the Segata lab. Following this, we used the Segata Lab’s custom taxonomy mapping together with the sequences of the MAGs to build a custom Kraken2 database, and classified the shotgun sequencing reads from the Cameroon and US samples using this database. Our goal in this approach was to test whether the results of our analyses were broadly reproducible when the MAGs were used as the reference, as an exhaustive comparison of results obtained using different reference databases or computational methods is beyond the scope of this manuscript.

This approach classified the majority of our shotgun sequencing reads (~60-85% in both groups). Importantly, the results of this analysis were concordant with our other analyses (shotgun metagenomic data using the KrakenUniq database built from nt, and our 16S V4 sequencing results). Like these other analyses, using the kraken2 classifications obtained from the MAG sequence database demonstrated the predominance of *Prevotellaceae* in the Cameroonian fecal samples, contrasted with the dominance of *Bacteroidaceae* in the US samples (Sup. Fig. 9). Thus we think that KrakenUniq with the nt reference database is an appropriate approach for the analyses conducted in this paper.

We have updated the Discussion section of the paper to reflect our comparisons:

“As an additional point of comparison, we queried our reads against a representative set of ~5000 metagenome-assembled genomes (MAGs) from Pasolli et al. (2019) [10], and found that the results of this analysis were concordant with those above (Sup. Fig. 9).”

We have further updated the Methods to reflect how the KrakenUniq-MAGs comparison was performed:

“As further validation, we compared percentages of reads that could be classified to the family level for shotgun metagenomic data KrakenUniq with metagenome-assembled genomes (MAGs) taxonomic information. We obtained the sequences for the representative MAGs from (http://segatalab.cibio.unitn.it/data/Pasolli_et_al.html) the taxonomic information published in the metadata and corresponding NCBI Taxonomy identifiers [12] kindly provided by members of the Segata lab, we constructed a custom Kraken2 database [13], and classified the quality-controlled metagenomic reads described above using kraken2 as run by version 3.0 of the Sunbeam pipeline [14]. Downstream

visualization was performed using R and packages in the tidyverse; including ggplot2 [15–17].”

Finally, we have added an additional supplemental figure, Sup. Fig. 9, showing family level relative abundances from metagenomic sequences classified using the Kraken2 build of representative MAGs between HMP and Cameroon population:

Reviewer 1, Major Comment #5: I recommend that the authors be careful not to conflate correlation with causation - for example, "our sample set also allows us to ask how the different baseline microbiotas in rural populations are impacted by colonization with multiple parasites". Another example, " Infection with ANTS parasites, in turn, was associated with elevated levels of TH1, TH2, and proinflammatory cytokines, indicating engagement of multiple immune mechanisms." And yet another example, "These data document transkingdom interactions" - I'm not sure that "interactions" are actually documented, per se (though I acknowledge that the last example is a minor point).

Author Response: We have amended phrasing throughout the paper to separate correlation and causality. Examples include:

“We assessed these data for correlations between abundances of eukaryotic protists and prokaryotes (transkingdom correlations)”

“Infection with ANTS parasites, in turn, was associated with elevated levels of TH1, TH2, and proinflammatory cytokines, indicating associations with multiple immune mechanisms.”

“These data document transkingdom correlations.”

“Our sample set also allows us to ask how the different baseline microbiotas in rural populations are associated with colonization with multiple parasites.”

Reviewer 1 Major Comment #6: One of the most interesting findings in this manuscript, to me, is the finding that Bacteroidales is associated with higher proportion of ANTS infection. It would be helpful if the authors speculated about the potential reason for this finding. This is particularly so as Bacteroidales appears to be negatively correlated with E. histolytica in humans and in murine models, helminths are associated with a reduction of Bacteroidales.

Author Response: We have modified our discussion on the association between *Bacteroidales* and ANTS infection to speculate further on the potential reasons for this below-

“Multiple RFC models testing different categories of ANTS parasite infection and IL-5 cytokine response indicated that *Bacteroidales*, a known occupant of intestinal mucosal

surfaces and mucin degrader, was an important predictive taxon. *Bacteroidales* was consistently found at elevated abundances in ANTS positive individuals in comparison to ANTS negative individuals in this study. Previously, *Bacteroidales* has been found in lower abundances in the guts of humans infected with *Entamoeba histolytica* [7]. In murine models, infection with helminths led to the reduction of *Bacteroidales* and the concurrent expansion of *Clostridiales* communities [18], which was hypothesized to stimulate an anti-inflammatory response (increased IL-5 and IL-13) in the host. *Bacteroidales* have been shown in multiple studies to modulate intestinal and immune functions in the host [19–21]; here, *Bacteroidales* may be found in higher abundances due to possible direct interactions with ANTS parasites, through indirect means via immunomodulatory signaling as a result of ANTS infection, or due to an increase in mucin production consequent to nematode-induced infection [22].”

Reviewer 1, Major Comment #7. One major missing piece of metadata is diet - do the authors have any data on this important factor?

Author Response: Given the volume of participants and the logistics of working in the field, we were unable to do individual dietary records during the sample collection in Cameroon. However, their diets tend to be homogenous within each population. We did have access to population-based nutritional questionnaires, which were conducted at sampling sites within the three different administrative regions (South, East, and Northwest) at the time of participant recruitment and sample collection. Per Reviewer 1's comments, we have expanded the dietary description given in the methods and included the nutritional questionnaire as an additional supplemental table (new Sup. Table 2). We find that our dietary results are concordant with prior research on the diets of our sampled populations [6,7,23], and now use these data more fully in the interpretation.

We have added the following text to the Participant Details section of the Methods:

“For the Cameroon populations included in this study, we were able to obtain nutrition questionnaires that included types of food grown, foraged or hunted, sold, given to livestock, and consumed, with general frequencies (daily, weekly) of food consumption (Sup. Table 2). We also note what sources of water are used, where food is prepared, and if there had been recent periods of food insecurity. Our results were consistent with prior research [6,7,23] indicating that Baka and Bagyeli have fiber-high diets, incorporate food from foraging and hunting, and have some small-scale subsistence farming and raising of small livestock (primarily chickens). Bantu rely primarily on small-scale subsistence farming of grains and vegetables for their food and have a larger array of livestock used for meat and trade. The Fulani have similar diets to the Bantu with the notable exception that their diets include a substantial amount of dairy.”

Reviewer 1, Major Comment #8: Detection of parasites using microscopy is notoriously insensitive - what was the training of the individuals who evaluated the samples? Classically, ova and parasite tests are done on sequential stool samples to improve the sensitivity of the test.

Author Response: Our fecal and blood microscopy was performed by Dr. Valentine Ngum Ndze, who is currently the laboratory advisor for the USAID Infectious Disease Detection and Surveillance Project at African Society for Laboratory Medicine. We agree with the reviewer that microscopy detection of parasites is insensitive, and consequently we screened all samples after collection in the U.S. for parasite infection using qPCR, which is highly sensitive and specific. Although we report the high concordance between qPCR parasite detections with microscopy detections, all association testing with the microbiome and other analyses was performed using parasite positivity and parasite genomic copy number taken from qPCR results only. We have revised the text to clarify this point under the results as follows:

“Given the greater sensitivity, qPCR-confirmed parasite infections were used for subsequent analysis.”

And we have further revised the methods subheading “Quantitative PCR (qPCR)” as follows:

“All association testing with the microbiome and other analyses was performed using parasite positivity and parasite genomic copy number taken from qPCR results only.”

Reviewer 1, Major Comment #9: The inclusion of children (minimum age 4 in some categories) with adults would seem to be a substantial confounder. This is especially so in considering the lactase persistence analysis. Is there a reason that the authors favored combining children and adults into the same analysis? This is rather unusual.

Reviewer 1, Minor Comment #3: While interesting, the lactose nonpersistence vs. persistence section came across as a non sequitur

Reviewer 2, Minor Comment #1: One of the more surprising findings of the study was the fact that lactase persistence wasn't associated with microbiome composition in the Cameroonians, which is different from what is observed in European populations. It may be beneficial to include a few more details in the results section about these investigations. As written, it sounds like only overall composition was examined, and not associations between LP and specific microbes. Was the association between LP and Bifidobacteria tested (specifically, since that has what has been observed in European populations)? If an association wasn't observed (which I imagine it wasn't, or it would be included here), is the abundance of Bifidobacteria lower in these populations as compared to European populations? Additionally, the association between LP and Bifidobacteria is only apparent in European populations when there's milk in the diet (<https://www.ncbi.nlm.nih.gov/pubmed/27694959>). Did dairy consumption recorded for the different populations of this study? Is there potentially only an association in those individuals who are LP in the pastoralist population? Finally, given that there is a difference in galactose and lipid metabolism pathways in the pastoralists (which is a fascinating result), can

those genes be traced back to particular microbes using the metagenomic data (for example, do those genes increase in abundance because some group of microbes other than the Bifidobacteria increase in abundance in those individuals)?

Author Response: Both Reviewers 1 and 2 had helpful suggestions to improve our sections on lactose persistence/nonpersistence associations with the gut microbiome. Reviewer 1 asked about age. We found no significant differences when testing unweighted and weighted UniFrac distances in PERMANOVA with age (Sup. Tables S4 and S5); this is now mentioned in the revised paper. Per Reviewer 1's comments, we conducted an additional PERMANOVA using the following age bins: 4-10 (early childhood), 11-18 (juvenile), and 19+ (adult), however, neither test produced significant differences after multiple test correction. Median proportional abundances show no *Bifidobacteriaceae* > 0.000000000%.

We have added the following sentence to the results:

“We found no differences in weighted and unweighted microbiome UniFrac distances by sex or age using PERMANOVA.”

Reviewer 1 also commented that the lactose data came off as a *non sequitur*. We have thus added a new introduction to this section explaining the significance:

“The ability to digest milk as an adult is suggested to be an adaptive trait that can confer nutritional benefits and can provide a critical water source in arid regions [24]. Prior studies have contrasted industrialized populations with hunter-gatherer or agropastoralist populations, and only one has included African pastoralists known to consume high volumes of dairy. Here, we combine functional measures of the ability to breakdown milk sugar in Cameroonians, including a pastoral population

Reviewer 2 has listed several intriguing questions regarding our results on *Bifidobacteriaceae*. First, Reviewer 2 asks if we found an association between LP and Bifidobacteria in our Cameroonian populations- we did not. We had a total of 93 LNP adults (0.000000000% median proportional *Bifidobacteriaceae* abundance), 49 LP adults, (0.000000000% median proportional *Bifidobacteriaceae* abundance), 8 LNP juveniles (0.13% median proportional *Bifidobacteriaceae* abundance), 1 LP juvenile (not tested due to n=1), and no lactose tolerance test (LTT) data on younger children. Aggregating populations regardless of LP/LNP status, we found that the only Cameroon population with a non-zero median of *Bifidobacteriaceae* were the Bagyeli, with 0.02% relative abundance (compared to our US samples which had 0.5% relative abundance).

Reviewer 2 also asked how the abundance of *Bifidobacteria* in our populations compared to those of European populations. There is not, to our knowledge, a consensus on the relative abundance of *Bifidobacteria* across the gut microbiomes of healthy adult Europeans. Depending on the metric used, we have a zero or near-zero relative abundance of *Bifidobacteria*, so we can confidently say that this number is lower than the abundances reported in Japanese populations [25] and much lower than the amounts reported in the guts of pre-weaning infants (~20-80%) [26,27]. We added implications of this in the Discussion:

“This result implies that Cameroonians may not have the same adaptive response in the composition of their gut microbiota to dairy sugar as Europeans.”

“Several taxa with small relative abundances cumulatively produced the galactose metabolism pathway result in the Fulani, suggesting that their gut bacteria may act in consortium within an individual to catabolize dairy sugar. This is in contrast to a single taxon (i.e., *Bifidobacteria*) being attributed with much of the dairy sugar breakdown in the guts of Europeans.”

Reviewer 2 asked about the association of milk in the diets of our populations, and if tested for associations between lactose phenotypes within the Fulani pastoralists. Only the Fulani reported intake of dairy. Although all 23 Fulani contributed to shotgun metagenomic pathway analysis (and the result of galactose metabolism enrichment compared to other populations), only a subset also had LTTs performed (3 LP, 1 LNP), so we could not robustly look for an association of *Bacteroidaceae* with lactose phenotype in this group.

Finally, per the final portion of Reviewer 2’ s Minor Comment 1, we found that the most differential bacterium in the list produced from LefSe analysis of microbes whose genes increased in abundance in the galactose metabolism pathway for the Fulani was *Bacteroides vulgatus* (0.002% abundance). There were five lipid metabolism pathways that were significantly higher in the Fulani, so we have provided the most differentially abundant bacterium and their abundance in the Fulani below.

1. Glycerophospholipid metabolism: *Prevotella enoeca* (0.0003% abundance)
2. Sphingolipid metabolism: *Bacteroides vulgatus* (0.001% abundance)
3. Glycerolipid metabolism: *Bacteroides vulgatus* (0.0003% abundance)
4. Glycosphingolipid biosynthesis - ganglio series: *Bacteroides fragilis* (0.0001% abundance)
5. Glycosphingolipid biosynthesis - globo and isoglobo series Metabolism: *Bacteroides fragilis* (0.0002% abundance)

We have amended the text in the results on galactose and lipid pathways for the Fulani as follows below.

“The most highly abundant bacteria contributing to genes in these enriched pathways were *Bacteroides vulgatus* for galactose metabolism, and *Prevotella enoeca*, *Bacteroides fragilis*, and *Bacteroides vulgatus* for lipid metabolism.”

Reviewer 1, Comment #11: #. Schistosomiasis is a noted issue in Cameroon, and others (Ratard et al, PMID 2115306) have described a high rate of intestinal carriage of Schisto in Cameroon - is there a reason this was not detected in the studied cohort? Was it missing from the databases used for classification?

Author Response: We were surprised to find that all of our samples were negative for *S. mansoni* by light microscopy. We proceeded to test for *S. mansoni* by qPCR [28] on 100 samples. All were negative. Furthermore, the assembled reference genome for *S. mansoni* (https://www.ncbi.nlm.nih.gov/assembly/GCF_000237925.1) was included in our build of KrakenUniq from the standard nt database, as were other partial *S. mansoni* genomes, but we did not map any reads from our dataset to *S. mansoni*.

In checking the reported prevalence rates in Ratard et al. (1990)[29], we noted that by province/administrative region, the infection rates for *S. mansoni* were 0.6% for the South (5 infections out of 874 screened samples), 0.7% for the East (13/1762), and 0% for the Northwest (0/213). More recently, Tchuenté and colleagues (2012)[30] found that the Northwest was seeing a rise in *S. mansoni* infection rates (4.1%), while the South was seeing a decline (0.3%). Variation in regional rates of infection could be attributed to mass drug administration, promotion of hygiene and sanitation, and infection education efforts that have been underway since Cameroon began control activities in 2003 with the creation of the National Program for the Control of Schistosomiasis and Soil Transmitted Helminthiasis [30].

We have amended the methods to mention that we tested for *S. mansoni* but were unable to detect it, as follows:

“Samples were examined with and without iodine staining and visualized with standard light microscopy to identify visible gastrointestinal parasites or parasite ova, including hookworm (species indeterminate with light microscopy), amebiasis (*Entamoeba* spp.), giant roundworm (*Ascaris lumbricoides*), human whipworm (*Trichuris trichiura*), giardia (*Giardia* spp.), human roundworm (*Strongyloides stercoralis*), and flatworms (*Schistosoma mansoni*). We note that there were no positives for *S. mansoni* by light microscopy or by a qPCR test of 100 randomly selected samples and, therefore it was not included in further analyses.”

Reviewer 1, Minor Comment #1: What is a “longevity regulating pathway”? This needs some explaining/context.

Author Response: Longevity regulating pathways encompass a range of nutrient signaling pathways that are conserved through various organisms and are implicated in aging. These signaling pathways involve dietary restriction and genetic down regulation of nutrient-sensing, primarily through modification of insulin/insulin-like growth factor (IIS) and target-of-rapamycin (TOR) signaling. The full pathway description is visualized and described in detail in KEGG:

[https://www.genome.jp/kegg-bin/show_pathway?org_name=map&mapno=04213&mapscale=&show_description=show](https://www.genome.jp/kegg-bin/show_pathway?org_name=map&mapno=04213&mapscale=&show_description=show;) ;

References for this information are provided separately in KEGG at:

https://www.genome.jp/dbget-bin/www_bget?pathway+map04213

We have modified the text to better encapsulate this description as follows:

“However, ANTS positive individuals had enrichment for genes that play a role in bacterial purine and pyrimidine metabolism as well as nutrient signaling pathways implicated in aging (also known as “longevity regulating pathways”).”

Reviewer 2, Minor Comment #1: Perhaps a small or semantic point - but what separates a parasitic "infection" from "colonization". Given the high rate of colonization that was observed, is it possible that all colonization with candidate disease-causing parasites are not true "infections"? I realize this may be beyond the scope of the work presented, but given how interesting the parasite findings are, I think it is worthy of discussion and potentially having a parasitologist on the team comment on this.

Author Response: Defining colonization and infection is dependent on several factors including damage, host response, and specific characteristics of the hosts, microbes, and their interactions. For this paper, we have adapted the definitions for colonization and infection provided in Casadevall and Pirofski (2000) [31], who also provide an exhaustive review of prior definitions in the literature for these terms. Colonization here is used here to mean a state in which a microbe may be present within a host for an unspecified amount of time. Infections are the acquisition of a microbe inside the host, after which it replicates, and may cause disease. In the revised draft we have added more careful referencing on this point.

Reviewer 1, Minor Comment #4: *The speculation that “this higher diversity associated with helminth infection could result from parasite-induced rapid peristalsis...” This is intriguing, but would be stronger and more convincing if the association between parasites and peristalsis/epithelial shedding was cited.*

Author Response: Although a possibility, there is currently little evidence to date that rapid peristalsis is correlated with higher microbial diversity, and upon further consultation with the literature, we have modified this section to reflect a direct/indirect helminth immunomodulatory scenario backed up by multiple studies [32–34]:

“We speculate that this higher diversity associated with helminth infection could result from direct or indirect communication with other microbes and/or with immunomodulatory mechanisms (i.e., cytokines).”

Reviewer 1, Minor Comment #5: Re Entamoeba co-infection - are the authors sure that the findings of multiple taxa of Entamoeba being present are the result of co-infections as opposed to infection with a single intermediate organism for which there isn't yet a reference genome? What is the evidence for co-infection vs. this alternative model? I wonder whether co-detection of multiple species is a legitimate finding or if it is the result of "taxon splitting" by Kraken. One could use bracken to re-estimate species-level abundance (per the kraken documentation, it appears that bracken is compatible with Kraken-Uniq).

Author Response: The aim of this particular analysis was to identify if any of our metagenomic data contained species of *Entamoeba* known to infect humans with reference genomes in publicly available databases. The reviewer is correct that multiple infections could include a novel, intermediate form of *Entamoeba*, however, such a study would require considerable wet and dry side analyses to isolate, amplify, sequence, and analyze individual *Entamoeba* genomes, and is beyond the scope of this research. However, we employed rigorous criteria to ensure that we annotated our *Entamoeba* reads to known species, as detailed below.

For the *Entamoeba* co-infection analysis, we only used metagenomic data that was directly assigned at the species level to 5 species known to infect humans. KrakenUniq classifies metagenomic sequence reads by exact matching of short strings (default k-mer length of 31 nucleotides was used in this study). It further *estimates* how much of the genome is covered to discriminate between false and true positive hits. This is done by counting how many unique k-mers of a genome are covered by metagenomic reads within a sample. We checked whether increases in classified reads correspond to increases in unique k-mers (Total Reads:Total k-mers), as a proxy for increased genome coverage. Thus we required a minimum number of thirty unique hits and minimum confidence value ratio (<1) for each species to assign a positive hit. Because of these caveats, we did not evaluate relative abundance for each of these species, but rather looked at presence or absence, thus negating the need for a re-estimation of species-level abundances, per Reviewer 1's suggestion. Infection with a novel form of *Entamoeba* is entirely speculative at this point since we are limited in our ability to annotate something in our dataset as an *Entamoeba* if it doesn't have genomic sequences for reference in an existing database.

Reviewer 1, Minor Comment #6: While blood parasitemia (malaria) was investigated in this manuscript, it is not featured in most of the analyses that are presented, which makes it seem like a disconnected aspect of the manuscript.

Author Response: At the onset of this project, we broadly screened for fecal and blood parasites, and hypothesized that the most likely parasite to be correlated with the gut microbiome would be a gastrointestinal parasite given their shared environment with other gut microbiota. However, we did not want to rule out the potential for a blood parasite association with the gut microbiome composition. Thus, we included parasites from both tissues in this research.

We have modified the manuscript to incorporate this information under the sub-heading “Quantification of pathogens and their correlates with host physiology”:

“To compare microbiome-pathogen interactions in these Cameroonian populations, we acquired data on both intestinal and blood-borne pathogens.”

“The parasites that most frequently co-occurred were *A. lumbricoides*, *N. americanus*, *T. trichiura*, *S. stercoralis*, and *P. falciparum* (p-values < 0.05 by hypergeometric distribution) (Fig. 2B, Sup. Table 2). The four soil-transmitted helminth parasites are hereafter referred to as the “ANTS” group.”

Reviewer 1 is correct that aside from helminth infection, our highest prevalence pathogen was *Plasmodium falciparum* (Sup. Fig. 1). Indeed, we found significant cooccurrence with *P. falciparum* and *Trichuris trichiura* (Fig. 2B). However, *P. falciparum*, while significant in both unweighted and weighted PERMANOVA analysis, had lower R² values compared to ANTS parasites individually and as a group (Sup. Tables S3 and S4). Furthermore, *P. falciparum* RFC models underperformed the accuracy of RFC models using ANTS helminths individually and as a group (Sup. Table S6).

To better integrate this rationale into the paper, we have modified the following sections under the new results sub-heading “Association of microbiome structure and blood-borne pathogens” :

“*P. falciparum* infection also correlated with gut microbiome composition, and could be predicted from gut microbiome composition with ~73% accuracy. The top five taxa that best predicted *P. falciparum* infection status were *Bacteroidales*, *Roseburia faecis*, *Lachnospiraceae*, *Coprococcus*, and *Desulfovibrio* (Sup. Table S6). However, *P. falciparum* detection explained less variance in gut composition in PERMANOVA analysis than did ANTS parasites (Sup. Tables S4, S5), so we focused on the ANTS parasites in most of the analysis here.

Twenty-eight of the subjects were later found to be HIV-positive. Comparison of members of this group to HIV-negative individuals showed no distinction based on weighted and unweighted UniFrac distances in PERMANOVA analysis (Sup. Tables S4 and S5).”

We also conducted a correlation analysis between detection of *Plasmodium falciparum* with cytokine values, and now include this data as a new supplemental figure (Sup. Fig. 13D) and associated statistics (Sup. Table 13). We have amended the section “Comparison of microbial data to human immune responses reported by cytokine production” to include the following text:

“HIV and *P. falciparum* had positive, significant associations with proinflammatory cytokine TNF α (Spearman’s correlation coefficient, p-values < 0.01, Sup. Fig. 13C, Sup.

Table 11) and IL-10 (Spearman's correlation coefficient, p-values < 0.01, Sup. Fig. 13D, Sup. Table 13), respectively.”

Finally, we added a segment to the results section touching on the correlation between *P. falciparum* and IL-10:

“We found an additional modest but significant correlation between *P. falciparum* and IL-10. IL-10 is a potent anti-inflammatory cytokine that can ameliorate malaria pathology and secretes antibodies that can protect against malaria reinfection [35,36].”

Reviewer 1, Minor Comment #7: The authors suggest that longitudinal data would be helpful as a follow on, why? Is there a strong seasonality to diet and exposures in the Cameroonian individuals studied in this project? In comparison, the Hadza have a very strongly seasonal diet and lifestyle.

Author Response: The Cameroon populations included in this study were cross-sectionally sampled between January and July of 2015. Longitudinal sampling would allow us to 1) Control for intra-individual variation in the gut microbiome 2) Test for any differences between wet and dry seasons for parasite exposure, and 3) Test for mild effects in seasonality. This is now explained a bit further in the revised draft in the discussion section:

“Further studies would benefit from longitudinally sampling populations and integrating individualized dietary information to distinguish healthy host microbiome structure from parasitized states, and to test for association of microbial diversity with seasonality.”

Reviewer 1, Minor Comment #8: Please include details of the primers used for 16S sequencing.

Author Response: Sequences of the 16S primers are given in Sup. Table 1. Further details are provided in the Methods, subsection “Bacterial 16S rRNA Amplicon Sequencing,” which we have partially provided below:

“PCR reactions were performed on extracted fecal DNA in triplicate using Accuprime Pfx Supermix (Invitrogen; Carlsbad, CA) and barcoded composite primers with Illumina adapters to amplify the V4 region of the bacterial 16S rDNA genome following the methods of Kozich et al. (2013) [37] on a GeneAmp 9700 PCR System. Sequences of DNA primers used in this study are reported in Sup. Table 1. PCR conditions were as follows: 95°C for 2 min, followed by 30 cycles of 95°C for 20 sec., 55°C for 15 sec., 72°C for 5 min., and then a final elongation step at 72°C for 10 min.”

Reviewer 1, Minor Comment #9: What is "Sup.Table: Metadata"?

Author Response: We have edited all instances of this to “Sup. Table 1.”

Reviewer 1, Minor Comment #10: Presumably exclusion criteria for the baseline glucose was EITHER glc outside of 60-100 OR a diagnosis of Diabetes? Please clarify the writing of this sentence.

Author Response: We have edited this section to read as follows:

“Exclusion criteria included having a baseline glucose outside of 60-100 mg/dl or diabetes.”

Reviewer 1, Minor Comment #11: Perhaps I missed this, but how are the authors inferring genome size of Prevotella and Bacteroides using their shotgun data? I didn't see a section on genome assembly and binning, which presumably was used for this purpose? Also, assembly of 16S into the metagenome assembled genomes is notoriously difficult - thus, I'm not sure I'd make much of the finding around the "average 16S copies" in Prevotella vs. Bacteroides.

Author Response: The average genome sizes and 16S copy numbers for *Prevotella* and *Bacteroides* species were calculated based on reference genomes deposited in NCBI. The reviewer is correct that recovery of bacterial genomes and integration of 16S sequences into metagenome-assembled genomes was not performed here. We have modified the methods to explain this more clearly:

“We noted that average 16S copy number across *Prevotella* and *Bacteroides* genomes varies (average 16S copies across 24 species of *Prevotella* = 4, average across 24 species *Bacteroides* = 5.3) (<https://rrndb.umms.med.umich.edu/genomes/>). In our shotgun metagenomics data, the average size of *Bacteroides* genomes was 5.3 Mbp and the average size of *Prevotella* genomes was 2.7 Mbp. Larger genome size and more 16S copy numbers in *Bacteroides* could account for some of the variation we see in higher relative abundances of this taxon in 16S versus shotgun sequencing compared to *Prevotella*.”

Reviewer 1, Minor Comment #12: The Blastocystis paragraph in the results comes out of left field, and doesn't really flow with the narrative. Perhaps it is somewhat misplaced?

Author Response: Per the reviewer's comment, we have reorganized this section of the paper to better incorporate the few sentences mentioning *Blastocystis* in a more logical progression, as shown below:

“Taxa that had significantly higher abundances in *Entamoeba* negative (Ent-) individuals were *Flavobacterium magnum*, *Shigella dysenteriae* (*S. dysenteriae*), and *Anoxybacillus kamchatkensis*. Taxa that had significantly higher abundances in *Entamoeba* positive (Ent+) individuals were *Erysipelotrichaceae*, *Trueperella pyogenes*, *Staphylococcus aureus*, and *Blastocystis hominis*. Members of the *Erysipelotrichaceae* family have been associated specifically with *Entamoeba* infection in western lowland gorillas [38] and in humans [39]. Both *S. aureus* (associated with Ent+) and *S. dysenteriae* (associated with Ent-) can induce changes in *E. histolytica* virulence and host response through modification of *E. histolytica* surface lectin expression, adhesion, cytotoxicity, and proteolysis [40]. *Trueperella pyogenes* (Ent+), *Flavobacterium magnum* (Ent-), and *Anoxybacillus kamchatkensis* (Ent-) have not been associated before, to our knowledge, with *Entamoeba* infection status. Finally, *Blastocystis hominis* (*B. hominis*) is a unicellular protozoan found in human large intestines and stool at rates higher than any other parasite in non-industrialized countries [41,42]. Although *B. hominis* is usually considered a non-pathogenic commensal, and we had not detected it with microscopy, *B. hominis* has been noted to associate with increased diversity of human gut bacteria [43].

Reviewer 1, Minor Comment #13: Typo: “differentially enriched amount the three subsistence groups”

Author Response: We have modified this sentence as follows:

“Several KEGG classes were significantly differentially enriched among the three subsistence groups (Fig. 7A; Sup. Table. 7).”

Reviewer 2, Minor Comment #2: Ensure that the S of 16S is consistently capitalized throughout the manuscript.

Author response: We have insured that all instances of this are now capitalized.

Reviewer 2, Minor Comment #3: Legend of Figure 3F: Fill in the correct Supplemental Figure number (currently XX)

Author response: This supplemental figure number has been referenced in the caption for 3F as Supplemental Figure 10B.

References:

1. Jarvis JP, Scheinfeldt LB, Soi S, Lambert C, Omberg L, Ferwerda B, et al. Patterns of ancestry, signatures of natural selection, and genetic association with stature in Western African pygmies. *PLoS Genet.* 2012;8:e1002641.
2. Fan S, Kelly DE, Beltrame MH, Hansen MEB, Mallick S, Ranciaro A, et al. African evolutionary history inferred from whole genome sequence data of 44 indigenous African populations. *Genome Biology.* 2019;20:82.
3. Fagny M, Patin E, MacIsaac JL, Rotival M, Flutre T, Jones MJ, et al. The epigenomic landscape of African rainforest hunter-gatherers and farmers. *Nat Commun.* 2015;6:10047.
4. Patin E, Laval G, Barreiro LB, Salas A, Semino O, Santachiara-Benerecetti S, et al. Inferring the Demographic History of African Farmers and Pygmy Hunter–Gatherers Using a Multilocus Resequencing Data Set. *PLOS Genet.* 2009;5:e1000448.
5. Tishkoff SA, Reed FA, Friedlaender FR, Ehret C, Ranciaro A, Froment A, et al. The genetic structure and history of Africans and African Americans. *Science.* 2009;324:1035–44.
6. Gomez A, Petrzekova KJ, Burns MB, Yeoman CJ, Amato KR, Vlckova K, et al. Gut Microbiome of Coexisting BaAka Pygmies and Bantu Reflects Gradients of Traditional Subsistence Patterns. *Cell Reports [Internet].* 2016 [cited 2016 Mar 1];0. Available from: <http://www.cell.com/article/S2211124716300997/abstract>
7. Morton ER, Lynch J, Froment A, Lafosse S, Heyer E, Przeworski M, et al. Variation in Rural African Gut Microbiota Is Strongly Correlated with Colonization by *Entamoeba* and Subsistence. *PLOS Genet.* 2015;11:e1005658.
8. Wear EK, Wilbanks EG, Nelson CE, Carlson CA. Primer selection impacts specific population abundances but not community dynamics in a monthly time-series 16S rRNA gene amplicon analysis of coastal marine bacterioplankton. *Environ Microbiol.* 2018;20:2709–26.
9. Klindworth A, Pruesse E, Schweer T, Peplies J, Quast C, Horn M, et al. Evaluation of general 16S ribosomal RNA gene PCR primers for classical and next-generation sequencing-based diversity studies. *Nucleic Acids Res.* 2013;41:e1.
10. Pasolli E, Asnicar F, Manara S, Zolfo M, Karcher N, Armanini F, et al. Extensive Unexplored Human Microbiome Diversity Revealed by Over 150,000 Genomes from Metagenomes Spanning Age, Geography, and Lifestyle. *Cell.* 2019;176:649–662.e20.
11. Pasolli E, Asnicar F, Manara S, Zolfo M, Karcher N, Armanini F, et al. Extensive Unexplored Human Microbiome Diversity Revealed by Over 150,000 Genomes from Metagenomes Spanning Age, Geography, and Lifestyle. *Cell.* Elsevier Inc.; 2019;176:649–662.e20.
12. Federhen S. The NCBI Taxonomy database. *Nucleic Acids Research.* 2012;40:136–43.
13. Wood DE, Lu J, Langmead B. Improved metagenomic analysis with Kraken 2. *Genome Biology.* *Genome Biology;* 2019;20:1–13.
14. Clarke EL, Taylor LJ, Zhao C, Connell J, Lee J-J, Fett B, et al. Sunbeam: an extensible pipeline for analyzing metagenomic sequencing experiments. *Microbiome.* *Microbiome;* 2019;7:2–13.
15. Ihaka R, Gentleman R. R: A Language for Data Analysis and Graphics. 2012;5.
16. Wickham H. *ggplot2: Elegant Graphics for Data Analysis.* Springer-Verlag New York; 2016.
17. Wickham H, Averick M, Bryan J, Chang W, D’ L, McGowan A, et al. Welcome to the Tidyverse. *Journal of Open Source Software.* 2019;4:1686.

18. Ramanan D, Bowcutt R, Lee SC, Tang MS, Kurtz ZD, Ding Y, et al. Helminth infection promotes colonization resistance via type 2 immunity. *Science*. 2016;
19. Hooper LV, Stappenbeck TS, Hong CV, Gordon JI. Angiogenins: a new class of microbicidal proteins involved in innate immunity. *Nat Immunol*. 2003;4:269–73.
20. Zitomersky NL, Atkinson BJ, Franklin SW, Mitchell PD, Snapper SB, Comstock LE, et al. Characterization of Adherent Bacteroidales from Intestinal Biopsies of Children and Young Adults with Inflammatory Bowel Disease. *PLoS One* [Internet]. 2013 [cited 2020 Jan 27];8. Available from: <https://www.ncbi.nlm.nih.gov/pmc/articles/PMC3679120/>
21. Mazmanian SK, Liu CH, Tzianabos AO, Kasper DL. An immunomodulatory molecule of symbiotic bacteria directs maturation of the host immune system. *Cell*. 2005;122:107–18.
22. Grecnis RK. Immunity to helminths: resistance, regulation, and susceptibility to gastrointestinal nematodes. *Annu Rev Immunol*. 2015;33:201–25.
23. Afolayan AO, Ayeni FA, Moissl-Eichinger C, Gorkiewicz G, Halwachs B, Högenauer C. Impact of a Nomadic Pastoral Lifestyle on the Gut Microbiome in the Fulani Living in Nigeria. *Front Microbiol* [Internet]. 2019 [cited 2020 Jan 9];10. Available from: <https://www.frontiersin.org/articles/10.3389/fmicb.2019.02138/full#supplementary-material>
24. Ségurel L, Bon C. On the Evolution of Lactase Persistence in Humans. *Annual Review of Genomics and Human Genetics*. 2017;18:null.
25. Odamaki T, Kato K, Sugahara H, Hashikura N, Takahashi S, Xiao J-Z, et al. Age-related changes in gut microbiota composition from newborn to centenarian: a cross-sectional study. *BMC Microbiol*. 2016;16:90.
26. Lim ES, Zhou Y, Zhao G, Bauer IK, Droit L, Ndao IM, et al. Early life dynamics of the human gut virome and bacterial microbiome in infants. *Nat Med*. 2015;21:1228–34.
27. Baumann-Dudenhoeffer AM, D'Souza AW, Tarr PI, Warner BB, Dantas G. Infant diet and maternal gestational weight gain predict early metabolic maturation of gut microbiomes. *Nat Med*. 2018;24:1822–9.
28. ten Hove RJ, Verweij JJ, Vereecken K, Polman K, Dieye L, van Lieshout L. Multiplex real-time PCR for the detection and quantification of *Schistosoma mansoni* and *S. haematobium* infection in stool samples collected in northern Senegal. *Transactions of the Royal Society of Tropical Medicine and Hygiene*. 2008;102:179–85.
29. Ratard RC, Koueméni LE, Bessala MM, Ndamkou CN, Greer GJ, Spilisbury J, et al. Human schistosomiasis in Cameroon. I. Distribution of schistosomiasis. *Am J Trop Med Hyg*. 1990;42:561–72.
30. Tchuem Tchuenté L-A, Kamwa Ngassam RI, Sumo L, Ngassam P, Dongmo Noumedem C, Nzu DDL, et al. Mapping of Schistosomiasis and Soil-Transmitted Helminthiasis in the Regions of Centre, East and West Cameroon. *PLoS Negl Trop Dis*. 2012;6:e1553.
31. Casadevall A, Pirofski L. Host-Pathogen Interactions: Basic Concepts of Microbial Commensalism, Colonization, Infection, and Disease. *Infect Immun*. 2000;68:6511–8.
32. Zakeri A, Hansen EP, Andersen SD, Williams AR, Nejsum P. Immunomodulation by Helminths: Intracellular Pathways and Extracellular Vesicles. *Front Immunol* [Internet]. 2018 [cited 2020 Jan 30];9. Available from: <https://www.frontiersin.org/articles/10.3389/fimmu.2018.02349/full>
33. Maizels RM, Smits HH, McSorley HJ. Modulation of Host Immunity by Helminths: The Expanding Repertoire of Parasite Effector Molecules. *Immunity*. 2018;49:801–18.

34. Osborne LC, Monticelli LA, Nice TJ, Sutherland TE, Siracusa MC, Hepworth MR, et al. Virus-helminth coinfection reveals a microbiota-independent mechanism of immunomodulation. *Science*. 2014;345:578–82.
35. Kumar R, Ng S, Engwerda C. The Role of IL-10 in Malaria: A Double Edged Sword. *Front Immunol* [Internet]. 2019 [cited 2020 Feb 3];10. Available from: <https://www.frontiersin.org/articles/10.3389/fimmu.2019.00229/full>
36. Couper KN, Blount DG, Riley EM. IL-10: The Master Regulator of Immunity to Infection. *The Journal of Immunology*. 2008;180:5771–7.
37. Kozich JJ, Westcott SL, Baxter NT, Highlander SK, Schloss PD. Development of a dual-index sequencing strategy and curation pipeline for analyzing amplicon sequence data on the MiSeq Illumina sequencing platform. *Appl Environ Microbiol*. 2013;AEM.01043-13.
38. Vlčková K, Paččo B, Petrželková KJ, Modrý D, Todd A, Yeoman CJ, et al. Relationships Between Gastrointestinal Parasite Infections and the Fecal Microbiome in Free-Ranging Western Lowland Gorillas. *Front Microbiol* [Internet]. 2018 [cited 2019 Jun 27];9. Available from: <https://www.ncbi.nlm.nih.gov/pmc/articles/PMC6013710/>
39. Rosa BA, Supali T, Gankpala L, Djuardi Y, Sartono E, Zhou Y, et al. Differential human gut microbiome assemblages during soil-transmitted helminth infections in Indonesia and Liberia. *Microbiome* [Internet]. 2018 [cited 2018 May 10];6. Available from: <https://microbiomejournal.biomedcentral.com/articles/10.1186/s40168-018-0416-5>
40. Bär A-K, Phukan N, Pinheiro J, Simoes-Barbosa A. The Interplay of Host Microbiota and Parasitic Protozoans at Mucosal Interfaces: Implications for the Outcomes of Infections and Diseases. *PLoS Negl Trop Dis* [Internet]. 2015 [cited 2019 Jun 27];9. Available from: <https://www.ncbi.nlm.nih.gov/pmc/articles/PMC4684208/>
41. Jantermtor S, Pinlaor P, Sawadpanich K, Pinlaor S, Sangka A, Wilailuckana C, et al. Subtype identification of *Blastocystis* spp. isolated from patients in a major hospital in northeastern Thailand. *Parasitol Res*. 2013;112:1781–6.
42. Stenzel DJ, Boreham PF. *Blastocystis hominis* revisited. *Clin Microbiol Rev*. 1996;9:563–84.
43. Audebert C, Even G, Cian A, Blastocystis Investigation Group, Loywick A, Merlin S, et al. Colonization with the enteric protozoa *Blastocystis* is associated with increased diversity of human gut bacterial microbiota. *Sci Rep*. 2016;6:25255.

Second round of review

Reviewer 1

We appreciate the authors' carefully attention to the reviewer comments.

Overall, I think the manuscript is improved in terms of clarity and presentation.

My remaining minor concerns and suggestions are outlined below:

- 1) I continue to feel that the use of the term "infection" in the manuscript is misleading - just because a parasite/pathogen does not equate with actual infection (and could be considered asymptomatic colonization). Carefully reviewing the manuscript that the authors cite as a source for making this distinction, I disagree with their interpretation and thus think that the wording that they have selected should be carefully reconsidered. Perhaps discuss this wording with an infectious disease physician.
- 2) I don't feel strongly about this, but some may take umbrage to the use of the term "predict" in "Microbiome composition could be used to predict ANTS parasite coinfection" - this is an association, not a prediction; and again, I would refer to his as parasite presence or colonization as opposed to infection given the lack of information on symptoms.
- 3) It would be helpful if the reviewers include the discussion comparing/contrasting their results with that of Gomez et al in the actual discussion of the paper. This will help readers put the two studies in context.
- 4) Thanks to the authors for performing classifications against the MAG database. I find it interesting that there is a range of classifiability of reads. I wonder which communities have a lower rate of classifiability (closer to the 60% mark) - it would be interesting to make MAGs out of these samples (though understandably, is out of the scope of this paper).
- 5) Not all Bacteroidales are thought to degrade mucin, to my knowledge.

Below, we list specific responses to Reviewer 1's questions in the latest revision.

1) I continue to feel that the use of the term "infection" in the manuscript is misleading - just because a parasite/pathogen does not equate with actual infection (and could be considered asymptomatic colonization). Carefully reviewing the manuscript that the authors cite as a source for making this distinction, I disagree with their interpretation and thus think that the wording that they have selected should be carefully reconsidered. Perhaps discuss this wording with an infectious disease physician.

We have changed the term "infection" to "parasite occurrences," "parasite detections," "ANTS positivity," "detection," "ANTS positive," or "parasites" where it specifically references data generated in this research throughout the paper.

2) I don't feel strongly about this, but some may take umbrage to the use of the term "predict" in "Microbiome composition could be used to predict ANTS parasite coinfection" - this is an association, not a prediction; and again, I would refer to this as parasite presence or colonization as opposed to infection given the lack of information on symptoms.

We have amended the title to address Reviewer 1's concerns. The proposed title is now "Lifestyle and the presence of helminths is associated with gut microbiome composition in Cameroonians."

3) It would be helpful if the reviewers include the discussion comparing/contrasting their results with that of Gomez et al in the actual discussion of the paper. This will help readers put the two studies in context.

We have now included a modified version of our response to Reviewer 1's question on this subject in the first round of reviews within the paper's discussion section as follows:

"Two studies, Gomez et al. (2016) [4] and Morton et al. (2015) [9] have previously characterized the bacterial gut microbiomes of some central African rainforest hunter-gatherer populations and neighboring Bantu agropastoralists. Gomez et al. (2016) [4] hypothesized that enrichment of *Firmicutes* in the Bantu as compared to the Baka may associate with an increased ability to catabolize dietary energy. In contrast to this, we did not find an enrichment of *Firmicutes* in the Bantu as compared to the Baka. From 16S rRNA sequencing, we found that the median proportional abundance for *Firmicutes:Bacteroidetes* in our study was approximately 1:1 for both the Baka and the Bantu. The difference in *Firmicutes:Bacteroidetes* ratios between our study and Gomez et al. (2016) may be due in part to several factors, including distance from the sampling location in Gomez et al. (2016) to our nearest sampling location (>350 km apart) and diet, bias in assessment of microbial abundances introduced by different primers and amplicons between studies, and the comparatively larger number of individuals sampled in our study (117 Baka and 322 Bantu) compared to Gomez et al (2016) [4] (28 Baka and 29 Bantu)."

4) *Thanks to the authors for performing classifications against the MAG database. I find it interesting that there is a range of classifiability of reads. I wonder which communities have a lower rate of classifiability (closer to the 60% mark) - it would be interesting to make MAGs out of these samples (though understandably, is out of the scope of this paper).*

We also find this to be a fascinating area of investigation, but find it outside the scope of this research paper.

5) *Not all Bacteroidales are thought to degrade mucin, to my knowledge.*

To address variation in *Bacteroidales* catabolism, we have restructured the sentence mentioning mucin degradation thusly:

“Multiple RFC models testing different categories of ANTS parasite infection indicated that *Bacteroidales*, a known occupant of intestinal mucosal surfaces with mucin degrading species, was an important predictive taxon.”